

# Equilibrium fluctuations in maximally noisy extended quantum systems

**Michel Bauer[1,2], Denis Bernard[3⋆] and Tony Jin[3]**

**1** Institut de Physique Théorique de Saclay,
CEA-Saclay & CNRS, 91191 Gif-sur-Yvette, France.
**2** Département de mathématiques et applications, ENS-Paris, 75005 Paris, France.
**3** Laboratoire de Physique Théorique de l'Ecole Normale Supérieure, CNRS, ENS,
PSL University & Sorbonne Université, 75005 Paris, France.

⋆ denis.bernard@ens.fr

## Abstract

We introduce and study a class of models of free fermions hopping between neighbouring sites with random Brownian amplitudes. These simple models describe stochastic, diffusive, quantum, unitary dynamics. We focus on periodic boundary conditions and derive the complete stationary distribution of the system. It is proven that the generating function of the latter is provided by an integral with respect to the unitary Haar measure, known as the Harish-Chandra-Itzykson-Zuber integral in random matrix theory, which allows us to access all fluctuations of the system state. The steady state is characterized by non trivial correlations which have a topological nature. Diagrammatic tools appropriate for the study of these correlations are presented. In the thermodynamic large system size limit, the system approaches a non random, self averaging, equilibrium state plus occupancy and coherence fluctuations of magnitude scaling proportionally with the inverse of the square root of the volume. The large deviation function for those fluctuations is determined. Although decoherence is effective on the mean steady state, we observe that sub-leading fluctuating coherences are dynamically produced from the inhomogeneities of the initial occupancy profile.

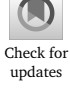

# 1 Introduction

Stochastic processes enter Quantum Mechanics from different corners : from a measurement perspective since, upon monitoring, quantum systems evolve randomly due to information readout and random back-action [1, 2], and from a statistical physics perspective, since the evolution of open quantum systems acquires some randomness through their interaction with external environments or reservoirs [3, 4]. The former lead to the notion of quantum trajectories [5–11] and its applications to quantum control [12–14]. The later may be modeled by coupling the quantum systems to series of noises, as exemplified by the Caldeira-Leggett or spin-boson models [15–17].

Putting aside the quantum nature of the environments leads to consider model systems interacting with classical reservoirs or noisy external fields. In the context of quantum many body systems, and especially quantum spin chains, the study of such models has recently been revitalized [18–20, 22–24] as a way to get a better understanding say of diffusive quantum transport, of entanglement production or of information spreading. They differ from quenched disordered dynamics because the noise is time-dependent and stochastic, but they share similarities with random quantum circuits recently considered [25–31]. Their dynamics are governed by unitary, but random, evolution operators $U_t$. Assuming the couplings to those reservoirs or external fields to be Markovian, the infinitesimal Hamiltonian generators $dH_t$ between time $t$ and $t + dt$, such that $U_{t+dt}U_t^\dagger = e^{-idH_t}$, can be written as $dH_t = H_0 \, dt + \sum_\alpha L_\alpha \, dB_t^\alpha$, where $H_0$ is some bare Hamiltonian and $L_\alpha$ a set of Hermitian operators to which the Brownian external fields $B_t^\alpha$ are coupled.

For the class of models we shall consider, the noisy contribution $\sum_\alpha L_\alpha \, dB_t^\alpha$ is *maximally noisy* in a sense to be made precise below which encodes for the ergodicity of the noisy flows, a property which can be mathematically formulated in terms of the Hörmander's theorem [32].

An iconic example of such models is the stochastic variant of the XX model describing fermions hopping from site to site on a 1D chain but with Brownian hopping amplitudes. We shall call this model the *quantum diffusive XX model*. Its dynamics is governed by the following

Hamiltonian generator, first introduced in [20],

$$dH_t = \sqrt{D} \sum_j \left( c_{j+1}^\dagger c_j \, dW_t^j + c_j^\dagger c_{j+1} \, d\overline{W}_t^j \right), \tag{1}$$

where $c_j$ and $c_j^\dagger$ are canonical fermionic operators, one pair for each site of the chain, with $\{c_j, c_k^\dagger\} = \delta_{j;k}$, and $W_t^j$ and $\overline{W}_t^j$ are pairs of complex conjugated Brownian motions, one pair for each edge along the chain, with quadratic variations $dW_t^j \, d\overline{W}_t^k = \delta^{j;k} \, dt$. It was shown that this model arises as the strong noise limit of the Heisenberg XX spin chain with dephasing noise, see Section 3.6 of [20]. If we start from a density matrix diagonal in the occupation number basis, the mean dynamics generated by this Hamiltonian can be mapped to the symmetric simple exclusion process [21]. It codes for a diffusive evolution of the number operators $\hat{n}_j = c_j^\dagger c_j$,

$$d\hat{n}_j = D \left( \Delta^{\mathrm{dis}} \hat{n} \right)_j \, dt + [\mathrm{Qu} - \mathrm{noise}],$$

with $\Delta^{\mathrm{dis}}$ the discrete Laplacian $(\Delta^{\mathrm{dis}} \hat{n})_j = \hat{n}_{j+1} - 2\hat{n}_j + \hat{n}_{j-1}$ and $[\mathrm{Qu} - \mathrm{noise}]$ some operator valued quantum noise. The parameter $D$ plays the role of the diffusion constant. The Hamiltonian generator (1) specifies stochastic flows on the fermion Fock space and we are going to show that the induced dynamics on the one-particle sector is maximally noisy in the sense alluded to above. This model is therefore (one of) the simplest model of quantum, stochastic, diffusion.

While most studies of open quantum systems [3,4], in contact with environments, focus on their mean behaviors by integrating the reservoir degrees of freedom, stochastic models such as the quantum diffusive XX model or its extensions allow to have access to the fluctuations of the system states or of quantum expectation values of series of observables. These fluctuations, which originate from the stochastic noise acting on the system, should not be confused with those arising from the quantum nature of the system, for which large deviation functions can be computed from the master equation alone [33,34].

The aim of the following is to present an exact description of the steady statistics, reached at large time, of the quantum states of the quantum diffusive XX model (1). Assuming the systems to be interacting with the noise but not driven out of equilibrium via contacts with external leads, we shall prove that such steady equilibrium statistics is universally described by a generating function simply represented in terms of the so-called Harish-Chandra-Itzykson-Zuber integral [36, 37] known in random matrix theory, as explained in the Proposition 1. This universality is an echo of the maximality of the noise in the one-particle sector and thus a consequence of the ergodicity of the flows the noise generates.

Furthermore, for infinitely large system size, the steady state is expected to be a non random, self averaging, state $\rho_{\mathrm{eq}}$ which is at equilibrium under the conditions we assumed. The results of Proposition 1 give access to the finite volume fluctuations $\delta\rho$ which, as we shall explain, scale proportionally to the inverse of the square root of the volume of the system, as expected. While all coherences are absent from the mean equilibrium state $\rho_{\mathrm{eq}}$ – because of the proliferation of incoherent interferences mediated by the environmental noise – they are present in the subleading fluctuations $\delta\rho$. Remarkably, these fluctuating, subleading, coherences manifest themselves in the large time asymptotic fluctuating state $\delta\rho$, even though they were absent from the initial system state. They are generated by the stochastic dynamics, a statement which may sound paradoxical – as noise is usually believed to break coherences – but which we shall make explicit in the following.

This article is organized as follows. In Section 2 we give a first hint on equilibrium properties of the model we are considering by deriving correlations functions to first few orders. In section 3, we derive the full stationary distribution non-pertubatively. In section 4, we present

two possible large systems size scalings of our stationary distribution and show that one of them is described by a large deviation function. In section 5, we explain in details the rules of the diagrammatic representation introduced in section 2. Finally, section 6 is devoted to a brief discussion on possible generalizations and possible future directions are pointed out. Some technical results are given in the appendices.

## 2 Low order correlations in the quantum diffusive XX model

We consider the quantum diffusive XX model on a ring with $L$ sites, with periodic boundary conditions – i.e. no open boundary conditions and no contact with external leads. We expect that, at large time, the system reaches an equilibrium steady state plus fluctuations. The aim of the two following Sections is to make this statement precise and to get a handle on these fluctuations.

Because the Hamiltonian generator (1) is quadratic in the fermion operators, the quantum diffusive XX dynamics can be solved with density matrices $\rho_t$ which are exponentials of quadratic forms in the fermion operators. We take $\rho_t = Z_t^{-1} \exp(c^\dagger M_t c)$ with $M_t$ a time dependent $L \times L$ Hermitian matrix and $Z_t$ the normalization partition function $Z_t = \text{Tr}(e^{c^\dagger M_t c}) = \det(1 + e^{M_t})$. This density matrix is parameterized either by the quadratic form matrix $M_t$ or by the two point function matrix $G_t$, with entries $(G_t)_{ij} = \text{Tr}(\rho_t c_j^\dagger c_i)$. All higher order quantum expectations can be derived from $G$ via the Wick's theorem. The system dynamics is then reduced from the fermionic Fock space, of dimension $2^L$, down to the one-particle Hilbert space, of dimension $L$, with $G_{t+dt} = e^{-idh_t} G_t e^{idh_t}$, or equivalently [1]

$$dG_t = -\frac{1}{2}[dh_t[dh_t, G_t]] - i[dh_t, G_t],$$

with one-particle Hamiltonian generator $dh_t$ given by,

$$dh_t = \sqrt{D} \sum_j \left( E_{j+1;j} \, dW_t^j + E_{j;j+1} \, d\overline{W}_t^j \right), \tag{2}$$

where $E_{j;k} := |j\rangle\langle k|$, $j,k \in [1,L]$ is the elementary $L \times L$ matrices, so $(E_{j;k})_{i;i'} = \delta_{i;j}\delta_{k;i'}$.

It is worth writing explicitly the stochastic equations of motion, which are SDE's, satisfied by the matrix of two-point function $G_t$, with $j \neq i$,

$$dG_{ii} = D(\Delta^{\text{dis}}G)_{ii}dt + i\sqrt{D}\left(G_{i;i-1}d\overline{W}^{i-1} + G_{i;i+1}dW^i - G_{i-1;i}d\overline{W}^{i-1} - G_{i+1;i}d\overline{W}^i\right), \tag{3}$$

$$dG_{ij} = -2D\,G_{ij}dt + i\sqrt{D}\left(G_{i;j-1}d\overline{W}^{j-1} + G_{i;j+1}dW^j - G_{i-1;j}d\overline{W}^{i-1} - G_{i+1;j}d\overline{W}^i\right), \tag{4}$$

with $(\Delta^{\text{dis}}G)_{ii} = G_{i+1;i+1} - 2G_{i;i} + G_{i-1;i-1}$. Recall that $G_{ii} = \text{Tr}(\rho_t \hat{n}_i)$ are the quantum mean occupation numbers. Let us denote them $n_i$ (i.e. $n_i = G_{ii}$). The first equation (3) codes for stochastic diffusion. In particular the mean occupation numbers $\mathbb{E}[n_i]$ diffuse according the heat equation, $d\mathbb{E}[n_i] = D\,\mathbb{E}[(\Delta^{\text{dis}}n)_i]dt$, and they attain a uniform profile at large time for periodic boundary conditions. The second equation (4) codes for decoherence and the mean off-diagonal elements $\mathbb{E}[G_{ij}]$ die off exponentially fast at large time.

We shall argue later that the distribution of the two point matrix reaches a stationary value at large time, so that the limit $\lim_{t\to\infty} \mathbb{E}[F(G_t)]$ exists for any (sufficiently regular) function $F$ and this defines an invariant measure $\mathbb{E}_\infty$ of the flows (2) on the one-particle sector. Because $G$ parameterizes the fermion density matrix this specifies an invariant measure of the flows (1) on the Fock space. The aim of the following is to determine it.

---

[1]We use the Itô convention to write stochastic differential equations (SDE).

The flows (2) are unitary flows and as such they preserve the spectrum of $G$, which is therefore completely specified by the spectrum of the initial condition $G_0$. In particular, all traces of powers of $G$ are non random constants of motion for the flows (2). Let $N_k = \text{tr}\, G^k$. The invariant measure $\mathbb{E}_\infty$ is parameterized by these conserved quantities.[2]

For $\mathbb{E}_\infty$ to be invariant means that $\mathbb{E}_\infty[F(G_t)]$ is time independent for any function $F$, which for instance can be chosen to be polynomial in $G$ of the form $\text{tr}\, A_1 G \cdots \text{tr}\, A_p G$ of arbitrary degree $p$ and with $A_1, \cdots, A_p$ generic $L \times L$ matrices. Demanding time independence of $\mathbb{E}_\infty[F(G_t)]$ yields constrains on those expectation values, which can be solved degree by degree because the evolution equations (3,4) are linear in $G$.

For degree one, we have to consider $\mathbb{E}_\infty[G_{ij}]$. As discussed above, we know that $\mathbb{E}_\infty[G_{ij}] = 0$ for $i \neq j$ and that $\mathbb{E}_\infty[n_i]$ is solution of $\mathbb{E}_\infty[(\Delta^{\text{dis}} n)_i] = 0$. Imposing periodic boundary conditions, as we assume, this implies that $\mathbb{E}_\infty[n_i]$ is uniform. The conserved quantity $N_1 = \text{tr}(G)$ then imposes that $\mathbb{E}_\infty[n_i] = N_1/L$, so that the mean steady state describes a uniform density as expected for such closed system.

For degree two, we have to consider $\mathbb{E}[G_{ij}G_{kl}]$. Using the dynamical equations (3,4) one can show that the only cases for which there is no exponential decay towards zero are $\{i = j, k = l\}$ and $\{i = l, j = k\}$. The former corresponds to $\mathbb{E}[n_i n_j]$ while the later to $\mathbb{E}[f_{ij}]$ where we introduced the notation $f_{ij} = G_{ij}G_{ji}$. They satisfy a closed set of equations valid in the steady state:

$$\mathbb{E}_\infty[(\Delta^{\text{dis}} n)_i \, n_j + n_i (\Delta^{\text{dis}} n)_j - 2((\delta_{i;j-1} - \delta_{i;j})f_{i;i+1} + (\delta_{i;j+1} - \delta_{i;j})f_{i;i-1})] = 0, \quad (5)$$

$$\mathbb{E}_\infty[f_{\Delta^{\text{dis}} k;k'} + f_{k;\Delta^{\text{dis}} k'} - 2(\delta_{k';k+1} n_k n_{k+1} + \delta_{k';k-1} n_{k-1} n_k)] = 0, \quad \text{for } k \neq k', \quad (6)$$

where we adopted the notation $f_{\Delta^{\text{dis}} k;k'} \equiv f_{k-1;k'} - 2f_{k;k'} + f_{k+1;k'}$. As can be seen by evaluating (5,6) for positions where the Kronecker's symbols are zero, the diffusive nature of these equations imposes that the expectations $\mathbb{E}_\infty[n_i n_j]$ (resp. $\mathbb{E}_\infty[f_{ij}]$) are all equal for $i \neq j$. The remaining terms are of the form $\mathbb{E}_\infty[n_i^2]$ which, by translational invariance, should also be site independent.

There is a useful graphical representation based on the analogy between the matrix $G$ and a propagator: each $G_{ij}$ is represented by an oriented arrow from site $i$ to site $j$. The systematics of such a representation is explained in 5, but it shouldn't be surprising to represent $\mathbb{E}_\infty[n_i] = \mathbb{E}_\infty[G_{ii}]$ by $[\!\text{⚬}\!]$, $\mathbb{E}_\infty[n_i n_j] = \mathbb{E}_\infty[G_{ii}G_{jj}]$ for $i \neq j$ by $[\!\text{⚬⚬}\!]$, $\mathbb{E}_\infty[f_{ij}] = \mathbb{E}_\infty[G_{ij}G_{ji}]$ for $i \neq j$ by $[\!\bigcirc\!]$ and $\mathbb{E}_\infty[n_i^2] = \mathbb{E}_\infty[G_{ii}^2]$ by $[\!\infty\!]$. The diagrammatic representation makes use of the site independence to exempt us with the explicit labeling of vertices, being understood that different vertices correspond to different indices. For the time being, the reader may view $[\cdots]$ as a simple delimiter, needed because $[\!\text{⚬⚬}\!] \neq [\!\text{⚬}\!]^2$. The contact terms in (5,6) impose all the same relation :

$$[\!\infty\!] = [\!\text{⚬⚬}\!] + [\!\bigcirc\!]. \quad (7)$$

The two conserved quantities $\text{tr}\, G^2 \equiv N_2$ and $(\text{tr}\, G)^2 = N_1^2$ yield two extra relations: $N_1^2 = L[\!\infty\!] + L(L-1)[\!\text{⚬⚬}\!]$ and $N_2 = L[\!\infty\!] + L(L-1)[\!\bigcirc\!]$. We have thus three independent equations (5,6,7) that fix the three unknowns :

$$[\!\infty\!] = \frac{N_1^2 + N_2}{L(L+1)}, \quad [\!\text{⚬⚬}\!] = \frac{LN_1^2 - N_2}{L(L^2-1)}, \quad [\!\bigcirc\!] = \frac{LN_2 - N_1^2}{L(L^2-1)}. \quad (8)$$

It is worth noticing that the initial information encoded in the off-diagonal terms $f_{ij}(t=0) = G_{ij}(t=0)G_{ji}(t=0)$, with $i \neq j$ is not dismissed in the steady state, since it has an impact on

---

[2]Throughout the paper, Tr denotes trace over the Fock quantum space, while tr denotes the trace over the one-particle sector.

the final values of $[\infty]$, $[\lozenge\lozenge]$, $[\bigcirc]$ via the $N_2$ dependence. Notice also that these correlation functions do not depend on the positions except whether the latter are in contact or not. This is what we mean by stating that the correlation functions are topological.

Similar derivations where carried out explicitly for correlations of order 3 and 4. Details for order 3 are given in the Appendix A.

## 3 Non perturbative stationary generating function

All polynomial correlation functions of the two point function matrix can be computed order by order following the strategy developed in the previous Section (even though the computations become more and more cumbersome). The aim of this section is to describe those correlation functions at all orders and to show that they have a universal character.

The derivation of this universal statistics relies on the observation that it is $U(L)$ invariant, as can indeed be checked on the first few orders using the formula derived above. For degree one, having a uniform mean density implies that $\mathbb{E}_\infty[\mathrm{tr}\,AG] = \overline{n}\,\mathrm{tr}\,A$ with $\overline{n} = N_1/L$ the density. For degree two, the topological nature of the expectations yields (cf. Appendix A)

$$
\begin{aligned}
\mathbb{E}_\infty[(\mathrm{tr}\,AG)^2] &= \left([\infty] - [\lozenge\lozenge] - [\bigcirc]\right)\mathrm{tr}\,A_d^2 + [\lozenge\lozenge]\,(\mathrm{tr}\,A)^2 + [\bigcirc]\,\mathrm{tr}\,A^2\\
&= [\lozenge\lozenge]\,(\mathrm{tr}\,A)^2 + [\bigcirc]\,\mathrm{tr}\,A^2,
\end{aligned}
$$

with $A_d$ the diagonal matrix with entries $\mathrm{diag}(A_{ii})$. We use the relation (7) for stationarity to go from the first to the second line and to cancel the term proportional to $\mathrm{tr}\,A_d^2$ which is not $U(L)$ invariant. Thus the steady state condition (7) imposes the $U(L)$ invariance of the expectations $\mathbb{E}_\infty[(\mathrm{tr}\,AG)^q]$, at least for low values of the order $q$. See Appendix A for a check at order 3. We claim, and shall prove, that this holds at any order so that the invariant measure only depends on the spectrum of the initial condition $G_0$. Let us introduce the generating function

$$
Z(A) := \mathbb{E}_\infty\left[e^{\mathrm{tr}\,AG}\right] = \sum_{q\geq 0}\frac{1}{q!}\mathbb{E}_\infty\left[(\mathrm{tr}\,AG)^q\right], \tag{9}
$$

depending on a generic $L \times L$ matrix $A$. It is the generating function of the correlation functions of $G$: Taking multiple derivatives of $Z(A)$ with respect to $A$ yields the multiple correlation functions of the matrix elements of $G$. Since $G$ is Hermitian we may restrict ourselves in the following to $A$ Hermitian (or anti-Hermitian to ensure convergence of the expectation (9)). We have the following

**Proposition 1**
Let $V_t$, $t \geq 0$, be the diffusion on $SU(L)$ defined by $V_{t+dt} = e^{-idh_t}V_t$ with

$$
dh_t = \sqrt{D}\sum_j(E_{j+1;j}\,dW_t^j + E_{j;j+1}\,d\overline{W}_t^j).
$$

*For any deterministic initial condition $G_0$, let $G_t$ be the process defined by $G_t := V_t G_0 V_t^{-1}$.*
*(i) The law of this process converges at $t \to \infty$ to the invariant measure $\mathbb{E}_\infty$ which satisfies the following properties :*
*(ii) It is $U(L)$ invariant, in the sense that $Z(A) = Z(VAV^\dagger)$ for any $V \in U(L)$.*
*(iii) Its generating function $Z(A)$ can be represented in terms of the so-called Harish-Chandra-Itzykson-Zuber integral on the special unitary group $SU(L)$ with respect to the invariant Haar measure (normalized to unit volume), namely*

$$
Z(A) = \int_{U(L)}d\eta(V)\,e^{\mathrm{tr}\,AV^\dagger G_0 V} = \left(\prod_{k=1}^{L-1}k!\right)\frac{\det\left(e^{a_i g_j}\right)_{i,j=1}^L}{\Delta(a)\Delta(g)}, \tag{10}
$$

*where $\eta$ is the Haar measure, $(a_i)_{i=1}^L$ and $(g_i)_{i=1}^L$ are the spectrum of A and $G_0$ respectively and $\Delta(a)$ (resp. $\Delta(g)$) are the Vandermonde determinants of A (resp. $G_0$), $\Delta(a) = \prod_{i<j}(a_i - a_j)$.*

We can translate this proposition as a statement about the fermionic density matrix:

**Corollary 1**
*The ensemble of fermionic density matrices $\rho = Z^{-1} \exp(c^\dagger M c)$ with M random $L \times L$ matrices and $Z = \det(1 + e^M)$, have a distribution stationary with respect to the dynamics of the quantum diffusive XX model (1) if M is picked such that $G = (1 + e^{-M})^{-1}$ is distributed with measure $\mathbb{E}_\infty$.*

Any $U(L)$ invariant measure is an invariant measure for unitary flows and in particular for the flows generated by (2). Thus proving the proposition amounts to show that the flow converges at infinite time to the $U(L)$ invariant measure which is unique and given by (10). Once the $U(L)$ invariance is established, the formula (10) follows from simple manipulation. The integral $\int d\eta(V) e^{\mathrm{tr} AV^\dagger G_0 V}$ is the Harish-Chandra-Itzykson-Zuber integal [36,37].

The key point in proving that the invariant measure $\mathbb{E}_\infty$ is $U(L)$ invariant relies on the fact that, because they form a system of simple root generators, the matrices $E_{j;j+1}$ and $E_{j+1;j}$ generate the Lie algebra $su(L)$ so that iterated products of the form

$$e^{-idh_{t_1}} e^{-idh_{t_2}} \cdots e^{-idh_{t_n}},$$

for any collection of time increments $dt_k$, cover densely the group $SU(L)$. This is in the spirit of Hörmander's theorem for hypo-ellipticity of Fokker-Planck operators [32]. It means that the dynamics generated by successive iterations of the infinitesimal group elements $e^{-idh_t}$ is ergodic enough to cover the group $SU(L)$. This is what it means to be *maximally noisy*.

Notice however that the noise as specified in the original model (1) is not ergodic on the unitary group of the fermionic Fock space (of dimension $2^L$) but only in the unitary group of the one-particle subspace (of dimension $L$). In other word, if one wants to be more precise, the noise in (1) is *one-particle maximally noisy*.

The detail of the proof is given in the Appendix B.

The irreducible representations of the group $SU(L)$ can be indexed by Young tableaux $Y$. The generating function $Z(A)$ can be expanded in characters of the unitary group in the form [37]:

$$Z(A) = \sum_Y \frac{1}{m(Y)!} \frac{\sigma_Y}{d_Y} \chi_Y(A)\chi_Y(G_0), \tag{11}$$

where $m(Y)$ is the number of boxes in $Y$, $\sigma_Y$ the dimension of the representation of the permutation group $S_{m(Y)}$ associated to $Y$ and $d_Y$, $\chi_Y(A)$ are respectively the dimension and the character of the representation of $SU(L)$ indexed by $Y$. This sum is graded because the character $\chi_Y(A)$ are polynomials in $A$ of degree $m(Y)$. The first few terms are :

$$Z(A) = 1 + \frac{1}{L}\square(A)\square(G_0) + \frac{1}{2}\left(\frac{2}{L(L+1)}\square\square(A)\square\square(G_0) + \frac{2}{L(L-1)}\begin{smallmatrix}\square\\\square\end{smallmatrix}(A)\begin{smallmatrix}\square\\\square\end{smallmatrix}(G_0)\right) + \dots. \tag{12}$$

Explicitly,

$$Z(A) = 1 + \frac{N_1}{L}\,\mathrm{tr} A + \frac{N_1^2 + N_2}{4L(L+1)}((\mathrm{tr} A)^2 + \mathrm{tr} A^2) + \frac{N_1^2 - N_2}{4L(L-1)}((\mathrm{tr} A)^2 - \mathrm{tr} A^2) + \cdots$$

$$= 1 + \frac{N_1}{L}\,\mathrm{tr} A + \frac{1}{2}\left((\mathrm{tr} A)^2 \frac{LN_1^2 - N_2}{L(L^2-1)} + \mathrm{tr} A^2 \frac{N_2 L - N_1^2}{L(L^2-1)}\right) + \cdots$$

$$= 1 + [\lozenge]\,\mathrm{tr} A + \frac{1}{2}((\mathrm{tr} A)^2 [\lozenge\lozenge] + \mathrm{tr} A^2 [\bigcirc]) + \cdots,$$

which coincides with the result obtained by the perturbative treatment. The third order terms are computed in Appendix A.

Though the Harish-Chandra-Itzykson-Zuber formula is compact and elegant, the presence of Vandermonde determinants in the denominator, which must cancel out, makes explicit computations difficult. For instance, taking for $A$ a diagonal matrix with a single non zero element requires a limiting procedure because the spectrum such an $A$ is highly degenerate. On the other hand, this describes the one site statistics of the mean particle number, which is a very basic observable. It turns out that this can be computed explicitly to all orders. Though the character expansion could surely be used to attack this question, we used a completely different approach based on invariant theory. We only quote the result here, relegating details to Appendix C: for $n = 0, 1, \cdots$, we have

$$\mathbb{E}_\infty[G_{ii}^n] = \frac{n!(L-1)!}{(L+n-1)!} \sum_{n_k, \sum_{k \geq 1} kn_k = n} \prod_k \frac{1}{n_k!} \left(\frac{N_k}{k}\right)^{n_k}.$$

As the random variable $G_{ii}^n$ takes its value in $[0, 1]$, its moments characterize the distribution completely. Thus we have obtained an explicit description of the statistical fluctuations of the particle number at one site at infinite time but finite $L$. Note the close connection between this formula and the cumulant expansion.

## 4 Large size systems

We investigate here two interesting large system size limits which one may consider depending on how the conserved quantities $N_k = \text{tr } G^k$ scale with the system size $L$. The motivation to consider these two regimes is explained in Appendix E.

The first case corresponds to conserved quantities extensive in the system size, so that $N_k/L = \rho_k$ with the densities $\rho_k$ finite as $L \to \infty$. An initial state corresponding to this scaling is for instance a factorized, diagonal, state with density matrix $\rho = \otimes_j r_j$ with $r_j$ diagonal in the fermion number basis. In particular, $\rho_1 = \frac{1}{L}\sum_j n_j =: \bar{n}$ and $\rho_2 = \frac{1}{L}\sum_j n_j^2 =: \overline{n^2}$ are the initial spatial mean occupancies and square occupancies, and from (8) we have

$$\mathbb{E}_\infty[n_i^2] = \overline{n^2} + O(1/L), \ \mathbb{E}_\infty[n_i n_j] = \overline{n^2} + O(1/L),$$
$$\mathbb{E}_\infty[|G_{ij}|^2] = \frac{(\Delta\bar{n})^2}{L} + O(1/L^2), \quad i \neq j,$$

with $(\Delta\bar{n})^2$ the variance of the initial occupancies $(\Delta\bar{n})^2 := \overline{n^2} - \bar{n}^2$. In particular, it entails for fluctuating non trivial coherences since, although vanishing in mean, the off-diagonal elements of $G$ have non zero variance: say $\mathbb{E}_\infty[|G_{ij}|^2]$ does not vanishes if the initial state is not factorized and uniform. We hence observe the interesting phenomena that fluctuating coherences are dynamically produced by the dynamics from the inhomogeneities in the density profile.

Within this scaling, the connected moments of order $q$ scale like $1/L^{q-1}$ to leading order. The generating function $W(A) = \log Z(A)$ is (Recall that $N_k/L = \rho_k$)

$$\begin{aligned} W(A) &= \rho_1 \text{tr} A + \frac{1}{2L}(\rho_2 - \rho_1^2)\,\text{tr} A^2 + \frac{1}{2L^2}(\rho_1^2 - \rho_2)(\text{tr} A)^2 \\ &\quad + \frac{1}{3L^2}(\rho_3 - 3\rho_2\rho_1 + 2\rho_1^3)\,\text{tr} A^3 + O(1/L^3). \end{aligned} \tag{13}$$

This corresponds to the following correlation functions (recall that $n_i = G_{ii}$ and $f_{ij} = |G_{ij}|^2$,

$i \neq j$):

$$
\mathbb{E}_\infty[n_1^{a_1} \cdots n_L^{a_L}] = \rho_1^{||a||} + \frac{1}{L}\Big(\sum_k \frac{a_k(a_k-1)}{2}\Big)\rho_1^{||a||-2}(\rho_2 - \rho_1^2) + O(1/L^2),
$$

$$
\mathbb{E}_\infty[f_{ij}\, n_1^{a_1} \cdots n_L^{a_L}] = \frac{1}{L}\rho_1^{||a||}(\rho_2 - \rho_1^2) + O(1/L^2), \quad \forall i \neq j,
$$

with $||a|| = \sum_k a_k$. Hence, to order $O(1/L^2)$, the only non-trivial variables are the occupancies $n_i = G_{ii}$ and the coherences $f_{ij} = |G_{ij}|^2$. All other products of $G$ are sub-leading in $1/L$. These formula have a simple interpretation: up to order $O(1/L^2)$ we can decompose the occupancies as $n_j = \rho_1 + \delta n_j$ where the $\delta n_j$'s are i.i.d. variables with zero mean and variance $\mathbb{E}_\infty[(\delta n_j)^2] = \frac{1}{L}(\rho_2 - \rho_1^2)$.

The interpretation is clear. To leading order in the system size, the two point function matrix $G$ converges to the non random, uniform, matrix $G_{\mathrm{eq}} = \rho_1 \mathbb{I}$, proportional to the identity, reflecting convergence toward equilibrium. There are sub-leading fluctuations, scaling proportionally with the inverse of the system size, so that we write

$$
G \simeq G_{\mathrm{eq}} + \frac{1}{\sqrt{L}}\delta G + O(1/L).
$$

The first term $G_{\mathrm{eq}}$ is the non random equilibrium matrix. The second one $\delta G$ fluctuates, according to (13).

We can better describe these fluctuations in terms of a large deviation function. Notice that $w(A) = \lim_{L\to\infty} \frac{1}{L}W(LA)$ is finite, order by order in power of $A$, with

$$
w(A) = \rho_1 \operatorname{tr} A + \frac{1}{2}(\rho_2 - \rho_1^2)\operatorname{tr} A^2 + \frac{1}{3}(\rho_3 - 3\rho_2\rho_1 + 2\rho_1^3)\operatorname{tr} A^3 + O(||A||^4). \tag{14}
$$

The first orders in the expansion (up to order 4) suggest that $w(A) = \sum_k \frac{1}{k}f_k \operatorname{tr} A^k$, where $f_k$ stands for the large $L$ scaling limit of $L^{k-1}\mathbb{E}_\infty[G_{i_1 i_2}G_{i_2 i_3}\cdots G_{i_k i_1}]$ with $i_1, \cdots, i_k$ all distinct. This can be understood intuitively as a consequence of the emergence, in the large $L$ scaling limit, of a thermodynamic (extensive) limit where correlation functions factorize over connected components (in the sense of the graphical representation). Note that this formula is also closely related to the outcome of the second scaling limit, to be introduced below.

Hence,

$$
\mathbb{E}_\infty\big[e^{L\operatorname{tr} AG}\big] \asymp_{L\to\infty} e^{L\,w(A)}.
$$

This equivalently means that the probability distribution for $G$ satisfies the large deviation principle. Namely, for $g$ any given $L \times L$ Hermitian matrix, we have

$$
\mathrm{Prob}_\infty(G = g) \asymp_{L\to\infty} e^{-L\,I(g)}, \tag{15}
$$

with rate function $I(g)$ the Legendre transform of $w(A)$. Indeed, assuming (15) we compute $\mathbb{E}_\infty\big[e^{L\operatorname{tr} AG}\big] \asymp \int dg\, e^{-L\,I(g)} e^{L\operatorname{tr} Ag} \asymp e^{Lw(A)}$ with $w(A) = \inf_g(\operatorname{tr} Ag - I(g))$. The series expansion for $I(g)$ is :

$$
I(g) = \frac{\operatorname{tr}(g - \rho_1\mathrm{Id})^2}{2(\rho_2 - \rho_1^2)} - \frac{\operatorname{tr}(g - \rho_1\mathrm{Id})^3}{3(\rho_2 - \rho_1^2)^3}(\rho_3 - 3\rho_2\rho_1 + 2\rho_1^3) + O(||g - \rho_1\mathrm{Id}||^4). \tag{16}
$$

To leading order, $G$ is Gaussian with mean $\rho_1\mathrm{Id}$ and variance $\big((\rho_2 - \rho_1^2)/L\big)^{1/2}$.

The second scaling we consider is when $N_k \propto L^k$. For instance, consider this time a factorized, diagonal state with density matrix $\rho = \otimes_j r_j$ with $r_j$ a matrix written in the fermion

number basis with $\frac{1}{2}$ for each entries. Again the tool we have used in this regime is invariant theory. We quote only the result here, relegating details to Appendix C:

$$\mathbb{E}_\infty\big[\frac{1}{n!}\operatorname{tr}(AG)^n\big] = \sum_{n_k,\,\sum_{k\geq 1} kn_k=n}\,\prod_k \frac{1}{n_k!}\left(\frac{N_k\operatorname{tr}A^k}{kL^k}\right)^{n_k} + o(L^0),$$

which can be resummed

$$\mathbb{E}_\infty[e^{\operatorname{tr}AG}] = e^{\sum_{k\geq 1}\frac{1}{k}\frac{N_k}{L^k}\operatorname{tr}A^k} + o(L^0),$$

with the proviso that $o(L^0)$ holds for a fixed order in the expansion of the exponential at large $L$ and with the assumption that traces of powers of $A$ remain finite at large $L$.

## 5  Diagrammatics

For any (gentle) function $f$ from the set of $L \times L$ matrices to an affine space, we shall denote by $[\cdots]$ the average defined by :

$$[f(G)] := \int_{U(L)} d\eta(V) f(VGV^{-1}).$$

Proposition 1 can alternatively be formulated as claiming that for any function of $G_t$:

$$\lim_{t\to+\infty}\mathbb{E}[f(G_t)] = [f(G_0)].$$

This notation for averages is identical to that introduced above when using the graphical representation, and indeed the graphs stand for certain functionals of $G$. The goal of this section is to describe the diagrammatic representation of averages in general.

We shall associate to each $2n$-plet $(i_1, j_1, i_2, j_2 \cdots, i_n, j_n) \in [1, L]^{2n}$ a diagram constructed as follows. The diagram has vertices labeled by $\{i_1, j_1, i_2, j_2 \cdots, i_n, j_n\}$. To make the point clear, this set may well have less than $2n$ elements, because repetitions do not count in the enumeration of a set. If $i$ and $j$ are two vertices, draw an oriented edge from $i$ to $j$ and label it with $m$ if $i_m = i$ and $j_m = j$. Thus the diagram has $n$ edges. It is clear that the diagram fully encodes for $(i_1, j_1, i_2, j_2 \cdots, i_n, j_n)$. For example ($n = 4, L \geq 5$),

$$(5, 3, 3, 2, 1, 1, 1, 1) \iff$$

Let us recall that multisets are kinds of sets (so in particular the order of enumeration does not matter), but for the fact that the same element can appear with a multiplicity. We shall need the $n$-multisets $\{\!\{i_i, \cdots, i_n\}\!\}$, and $\{\!\{i_i, \cdots, i_n\}\!\} = \{\!\{j_i, \cdots, j_n\}\!\}$ is exactly equivalent to the fact that there is (at least) one permutation $\sigma \in \mathfrak{S}_n$ such that $j_1 = i_{\sigma(1)}, \cdots, j_n = i_{\sigma(n)}$. Note that this implies that $\{i_i, \cdots, i_n\} = \{j_i, \cdots, j_n\}$ (an equality of sets).

Invariant theory, see Appendix C, allows to show that $\big[G^{\otimes n}\big]_{i_1 j_1, \cdots, i_n j_n} = 0$ (for every $L$) unless $\{\!\{i_i, \cdots, i_n\}\!\} = \{\!\{j_i, \cdots, j_n\}\!\}$, i.e unless the collection of $i$s counted with multiplicities and the collection of $j$s counted with multiplicities coincide. We say that the $2n$-plet $(i_1, j_1, i_2, j_2 \cdots, i_n, j_n)$ is admissible if $\{\!\{i_i, \cdots, i_n\}\!\} = \{\!\{j_i, \cdots, j_n\}\!\}$, and a diagram is admissible if the associated $2n$-plet is admissible. Thus we may restate our statement in $[O]_{i_1 j_1, \cdots, i_n j_n} = 0$

unless $(i_1, j_1, i_2, j_2 \cdots, i_n, j_n)$ is admissible. In the diagram associated to an arbitrary multiplet $(i_1, j_1, i_2, j_2 \cdots, i_n, j_n)$, the out-degree at a vertex, i.e. the number of edges leaving this vertex, is the multiplicity of this vertex in the multiset $\{\!\{i_i, \cdots, i_n\}\!\}$ and the in-degree at a vertex, i.e. the number of edges arriving at this vertex, is the multiplicity of this vertex in the multiset $\{\!\{j_i, \cdots, j_n\}\!\}$. Thus, a diagram is admissible if and only if the in and out degrees at each vertex are equal, which by a classical remark due to Euler means that the (strongly) connected components of the diagram can be traveled through in a closed journey by using every oriented edge once. The diagram above was clearly not admissible but its loopy component was. Here is an example of an admissible diagram ($n = 7, L \geq 6$):

$$(2, 1, 2, 4, 4, 2, 6, 4, 4, 1, 1, 2, 1, 6) \iff$$ 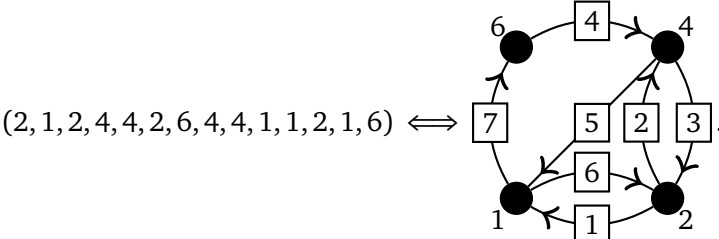

Given an admissible diagram $D$, we may define other closely related objects as follows. We call this the covering construction. At each vertex, pair the incoming edges with the outgoing edges, i.e associate to each incoming edge an outgoing edge (with two distinct edges arriving at a vertex being paired to two distinct edges leaving that vertex). The possibility of this pairing is nothing but the admissibility conditions. Suppose the diagram $D$ has $n$ edges. Associate to it a diagram with vertices labeled $1, 2, \cdots, n$ and for $l, m \in [1, n]$ draw an edge from vertex $l$ to vertex $m$ labeled $v$ if at vertex $v$ of $D$ the edge $l$ was incoming, the edge $m$ was outgoing and $l$ was paired to $m$. Then we say that $m$ is the successor of $l$ and $l$ the precursor of $m$. We call this new diagram a $*$-covering diagram of $D$.

Denoting by $d_v$ the (in or out) degree of vertex $v$ in a given admissible diagram $D$, the number of $*$-covering diagrams of $D$ is $\prod_{v \text{ of } D} d_v!$.

So our favorite example has 8 $*$-covering diagrams. Let us construct one. At vertex 1 pair edge 1 to edge 7 and edge 5 to edge 6. At vertex 2 pair edge 6 to edge 1 and edge 3 to edge 2. At vertex 4 pair edge 2 to edge 3 and edge 4 to edge 5. At vertex 6 pair edge 7 to edge 4. The resulting diagram is[3]:

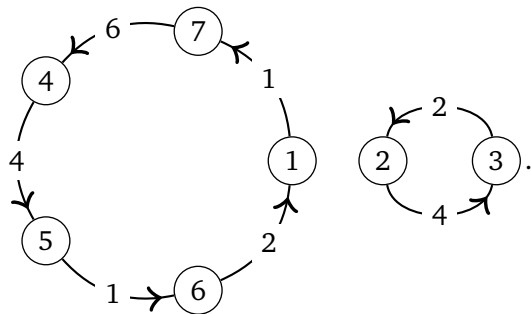

On the original admissible diagram $D$, and for each pairing, each edge has a unique successor and a unique precursor, so the edges of the $*$-covering diagram define a bijection on its set of vertices, i.e. on $[1, n]$. Thus the $*$-covering diagram defines an element of the permutation group $\mathfrak{S}_n$, say $\sigma$: each $*$-covering diagram is a collection of (decorated) cycles, as illustrated by our example. By construction the vertex label at the end of edge $m$ is the same as the vertex label at the beginning of edge $\sigma(m)$ i.e $i_{\sigma(m)} = j_m$ for $m = 1, \cdots, n$. Conversely, if $i_{\sigma(m)} = j_m$ for

---

[3]Note that we use a different convention for the original diagram and its derived diagrams to place the labels of the vertices and edges.

$m = 1, \cdots, n$ for some permutation $\sigma \in \mathfrak{S}_n$ and some $2n$-plet $(i_1, j_1, i_2, j_2 \cdots, i_n, j_n) \in [1, L]^{2n}$, we may decorate the cycle decomposition of $\sigma$ by labeling the edge joining $m$ and $\sigma(m)$ with $i_{\sigma(m)} = j_m$. Given an admissible diagram $D$ with $n$ edges and a permutation $\sigma \in \mathfrak{S}_n$, there is at most one way to decorate the edges of the cycle decomposition of $\sigma$ with the labels of the vertices of $D$ to get a $*$-covering diagram of $D$. So it is meaningful to say that a permutation covers $D$.

From a $*$-covering diagram, one can retrieve the original admissible diagram as follows. The first step is to rotate the edges of the $*$-covering diagram by half an edge (so that the roles of edges and vertices are exchanged, this rotation makes sense because the components of a covering graph are cycles). We obtain in this way what we call a covering diagram of $D$. The second step is to identify vertices of the covering diagram with carrying the same label. Again, let us illustrate the construction.

The edge rotation yields:

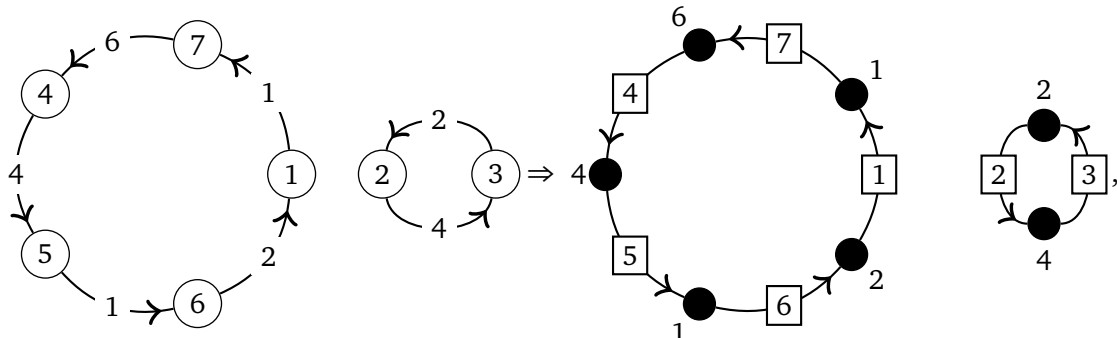

and then vertex identification yields the original admissible diagram:

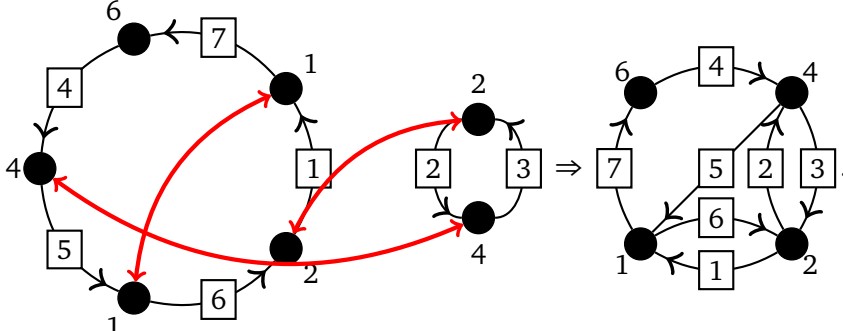

There are special admissible $2n$-plets $(i_1, j_1, i_2, j_2 \cdots, i_n, j_n)$ whose associated diagram $D$ has only one $(*\text{-})$covering: admissible diagrams where each vertex has a single incoming and a single outgoing edge. As there are $n$ edges, they must also be $n$ vertices, i.e. $i_1, \cdots, i_n$ must all be distinct. Let us call those admissible $2n$-plet $(i_1, j_1, i_2, j_2 \cdots, i_n, j_n)$ extremal. Then there is a unique permutation $\sigma \in \mathfrak{S}_n$ such that $i_{\sigma(m)} = j_m$ for $m = 1, \cdots, n$. Moreover, in this situation, the unique $*$-covering diagram of $D$ is indeed the diagram associated to the decomposition of $\sigma$ in cycles, decorated by the appropriate edge labels.

Notice that if $D$ is an arbitrary admissible diagram, edge rotation applied to any of its $*$-covering diagrams yields an extremal diagram from whom $D$ is recovered by identification of vertices carrying the same label: covering diagrams are always extremal diagrams.

Up to now, we have worked with graphs $D$ carrying labels on vertices and on edges. However, from the topological nature of averages, the vertex labels are irrelevant, it only matters that different vertices correspond to different points in $[1, L]$, so that vertex indices can safely be removed from the notation. Also, the tensor $G^{\otimes n}$ is symmetric under permutation of pairs

of indices, and this imples that edge labels are also irrelevant. Thus, as far as the computation of averages $[\cdots]$ are concerned, all labels can be removed. For instance, we may replace

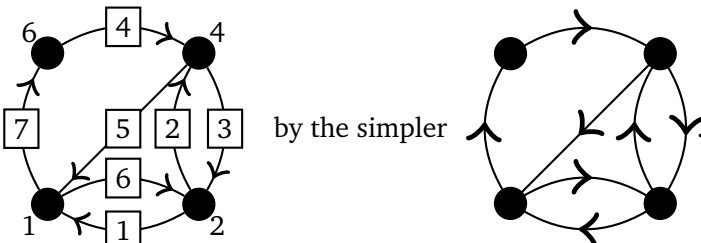

by the simpler

This is the rationale for the graphical representation (i.e. replacing the matrix element of $G^{\otimes n}$ by the associated unlabeled diagram with $n$ edges) of averages that we have used all along this work. The advantage of working wiht labeled graphs is that no multiplicities appear. This is not true when labels are removed. For instance the reader can easily work out that the 8 covering diagrams of our favorite example, which with labels present are all distinguishable, fall in only 4 classes after unlabeling:

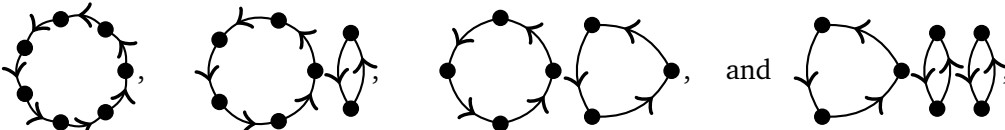

with multiplicities $3, 2, 2$ and $1$, leading to the expected $3 + 2 + 2 + 1 = 8$ covering diagrams.

Multiplicities are sometimes, but seldom in fact, related to symmetries and should not be confused with symmetry factors. Anyway, the algorithm to compute the $(*\text{-})$coverings of a diagram also work without labels, and in fact without labels there is no difference anymore between a $*$-covering and the associated covering obtained by edge rotation.

Extremal diagrams are the building blocks for averages. Indeed we have the following

**Lemma 2**
*Let $D$ be an unlabeled diagram associated to some matrix element of some power of $G$. Then $[D] = 0$, if $D$ isn't admissible, or*

$$[D] = \sum_{D' \text{ covering } D} m_{D'} [D'],$$

*where $m_{D'}$ is the number of times the extremal (unlabeled) diagram $D'$ appears as a covening of $D$.*

Thus, for our favorite example we have

$$\left[\vcenter{\hbox{}}\right] = 3 \left[\vcenter{\hbox{}}\right] + 2 \left[\vcenter{\hbox{}}\right] + 2 \left[\vcenter{\hbox{}}\right] + \left[\vcenter{\hbox{}}\right].$$

**Proof**
The formula is a direct consequence of its avatar for the labeled version of the diagrams, which for admissible diagrams reads

$$[D] = \sum_{D' \text{ covering } D} [D']$$

(multiplicity is 1 for each covering diagram). This labeled version is essentially a tautology once the invariants are known, see Appendix C for the details. □

The representation of arbitrary (admissible) diagram averages in terms of extremal diagram averages that this lemma provides is at the heart of the contact relations 7, 18 and their generalization to all orders. For instance the relation at order 2, $[\infty] = [\lozenge\lozenge] + [\bigcirc]$, is exactly the decomposition of $[\infty]$. This is also true of the first and third relations at order 3, $[\bigcirc\!\!\!-\!\!\circ] = [\lozenge] + [\lozenge\bigcirc]$ and $[\infty\!\!-\!\!\circ] = [\lozenge\lozenge\lozenge] + [\lozenge\bigcirc]$, while the second relation $[\ogreater] = [\infty\!\!-\!\!\circ] + 2[\bigcirc\!\!-\!\!\circ]$ is a consequence of the other two plus the decomposition $[\ogreater] = 2[\lozenge] + 3[\lozenge\bigcirc] + [\lozenge\lozenge\lozenge]$. Note that the second relation can also be interpreted as a kind of intermediate covering relation.

# 6 Generalization and conclusion

Let us now discuss how the previous statements can be generalized to stochastic random flows generated by Hamiltonian generators of the form

$$dH_t = H_0\,dt + \sqrt{D}\sum_j \big(c_{j+1}^\dagger c_j\,dW_t^j + c_j^\dagger c_{j+1}\,d\overline{W}_t^j\big), \tag{17}$$

with non trivial (preferably local) bare Hamiltonians $H_0$. We assume periodic boundary conditions.

The simplest case is $H_0 = \sum_j \mu_j c_j^\dagger c_j$ with chemical potential $\mu_j$. Such Hamiltonian preserves the form of the density matrix $\rho_t = Z_t^{-1}\exp(c^\dagger M_t c)$ and the flow induced on the one-particle sector is generated by :

$$dh_t = h_0 dt + \sqrt{D}\sum_j (E_{j+1;j}dW_t^j + E_{j;j+1}d\overline{W}_t^j),$$

with $h_0$ an $L \times L$ diagonal matrix with $\mu_j$ entries on the diagonal. We can absorb the $h_0$ dynamics by going to an interacting picture. Let $\tilde{G}_t = e^{ih_0 t}G_t e^{-ih_0 t}$, then

$$\tilde{G}_{t+dt} = e^{ih_0(t+dt)}e^{-idh_t}G_t\,e^{idh_t}e^{-ih_0(t+dt)} = e^{-id\tilde{h}_t}\,\tilde{G}_t\,e^{id\tilde{h}_t},$$

with $e^{-id\tilde{h}_t} = e^{ih_0(t+dt)}e^{-idh_t}e^{-ih_0 t}$, and

$$d\tilde{h}_t = \sqrt{D}\sum_j (E_{j+1;j}d\tilde{W}_t^j + E_{j;j+1}d\tilde{\overline{W}}_t^j),$$

where $d\tilde{W}_t^j = e^{i(\mu_{j+1}-\mu_j)t}dW_t^j$ and $d\tilde{\overline{W}}_t^j = e^{-i(\mu_{j+1}-\mu_j)t}d\overline{W}_t^j$. Let us remark that $d\tilde{W}_t^j$ and $d\tilde{\overline{W}}_t^j$ are complex conjugated Brownian motions with the same quadratic variations as before, $d\tilde{W}_t^j d\tilde{\overline{W}}_t^k = \delta^{j;k}dt$. Hence, the proof given in Section 3 applies and the stationary distribution for $\tilde{G}$ is again generated by the Harish-Chandra-Itzykson-Zuber integral. Because this measure is $U(L)$ invariant, the stationary distributions of $\tilde{G}$ and $G$ are identical, and thus independent of the chemical potentials $\mu_j$.

This last result indicates that even with disorder (e.g. by choosing the $\mu_j$'s to be random) a Brownian hopping destroys any signs of localization.

For the interacting case, take $H_0 = \sum_{k,l,m,n} V_{k,l,m,n}c_k^\dagger c_l c_m^\dagger c_n$. Since such evolution does not preserve the Gaussian form of the density matrix, we do not have an effective dynamics on the one-particle sector anymore. In the weak interaction regime –say there is some small scaling parameter $\lambda$ that weights the interacting Hamiltonian– we can do perturbation theory to get the first correction to the stationary distribution.

We now give a hint – but not a proof – why there exists an invariant measure independent of $\lambda$. Let us suppose that the stationary measure $\mathbb{E}_\infty^\lambda$ admits a perturbative expansion : $\mathbb{E}_\infty^\lambda[\bullet] = \mathbb{E}_\infty[\bullet] + \lambda \mathbb{E}_\infty^{(1)}[\bullet] + O(\lambda^2)$ with $\mathbb{E}_\infty$ the measure of the previous Sections whose support is on Gaussian states. Consider again the function $F_A(G) = \exp(\mathrm{tr}AG)$ with $G_{ji} = \mathrm{Tr}(\rho_t c_i^\dagger c_j)$. We can decompose its infinitesimal variation $dF_A(G)$ in two parts, the one generated by the free stochastic part $dF_A^{\mathrm{free}}(G)$ and the one generated by the interacting part $\lambda dF_A^{\mathrm{int}}(G)$ :

$$
\begin{aligned}
\mathbb{E}_\infty^\lambda[dF_A(G)] &= \lambda \mathbb{E}_\infty[dF_A^{\mathrm{int}}(G)] + \lambda \mathbb{E}_\infty^{(1)}[dF_A^{\mathrm{free}}(G)] + O(\lambda^2) \\
&= \lambda \Big( i\mathbb{E}_\infty[\mathrm{Tr}\big(\rho[H_0, c_l^\dagger c_k]\big)] \frac{\partial}{\partial G_{kl}} F_A(G)] + \mathbb{E}_\infty^{(1)}[dF_A^{\mathrm{free}}(G)]\Big) + O(\lambda^2).
\end{aligned}
$$

Demanding $\mathbb{E}_\infty^\lambda$ to be an invariant measure imposes that $\mathbb{E}_\infty^\lambda[dF_A(G)] = 0$. Using the Wick's theorem to evaluate $\mathrm{Tr}\big(\rho[H_0, c_l^\dagger c_k]\big)$, we can show that the first term of the previous line is zero. Hence, $\mathbb{E}_\infty^\lambda[dF_A(G)] = 0$ can be satisfied by choosing $\mathbb{E}_\infty^{(1)} = 0$. This indicates that the stationary distribution exposed in Section 3 might be an invariant measure even for non trivial $H_0$ but, of course, it is not a proof.

We expect the universality of the invariant measure to hold true for a large class of stochastic many-body quantum systems, such as the stochastic quantum spin chains considered in [20], as a consequence of a variant of Hörmander's theorem.

For the free case, it is clear that the results of previous Sections can be generalized in higher space dimension $D$ for similar models (for a given graph -say a $D$ dimensional lattice-, associate to each edge fermionic jumping operators with amplitudes given by local independent complex Brownians). It is also clear that the results of previous Sections can also apply to bosonic systems instead of fermionic ones.

A glance at the proof of the Proposition 1 reveals that it can be transferred to a large class of models. Consider quantum systems, defined over on Hilbert space $\mathcal{H}$, whose stochastic dynamics is generated by the noisy Hamiltonian $dH_t = H_0 \, dt + \sum_\alpha L_\alpha \, dB_t^\alpha$, as alluded to in the introduction. The proof of Proposition 1 is going to be applicable if iterative actions of the elementary unitaries $e^{-idH_t}$ are ergodic enough to cover the special unitary group of the system Hilbert space $SU(\mathcal{H})$. That property is going to hold if the operators $L_\alpha$ satisfy Hörmander criteria [32] which demands that their multiple commutators $[L_{\alpha_1}, [L_{\alpha_2}, [\cdots, L_{\alpha_M}] \cdots ]]$ span the Lie algebra of $SU(\mathcal{H})$, with or without the bare Hamiltonian $H_0 =: L_0$ included. This holds true if the set of $L_\alpha$'s contains at least a family of simple root generators of $su(\mathcal{H})$. In that case, the steady distribution of density matrix $\rho$ on $\mathcal{H}$ is such that its generating function $\mathbb{E}_\infty[e^{\mathrm{Tr}(M\rho)}]$, with $M \in GL(\mathcal{H})$, is the Harish-Chandra-Itzykson-Zuber integral (which of course depends on the spectrum of the initial density matrix $\rho_0$). The steady statistical behavior of these models will then share similarities with that of random unitary channels [25–28]. However the operators $L_\alpha$, and hence the noisy interactions, have to be non local on the chain to satisfy Hörmander's condition.

It is worth pointing out that, although fluctuations are encoded into the Harish-Chandra-Itzykson-Zuber integral, this last class of models and the quantum diffusive XX model (1) differs notably in the scaling behavior of their fluctuations. In both cases, the density matrix attains a non random equilibrium state $\rho_{\mathrm{eq}}$ at large volume or at large Hilbert space, so that

$$
\rho \simeq \rho_{\mathrm{eq}} + \delta\rho.
$$

But the fluctuating part $\delta\rho$ scales very differently in both cases : in the former class of models it scales inversely with the dimension of the Hilbert space as $1/\sqrt{\dim\mathcal{H}}$, which for a $q$-state spin model decreases exponentially with the system size as $q^{-\mathrm{vol.}/2}$, with 'vol.' the volume of

the system, whereas in the quantum diffusive XX model it scales inversely proportional to the volume as $1/\sqrt{\text{vol.}}$, which makes more physical sense for extended many-body systems.

The results described above open an avenue of explorations: by looking at the entangled characters of the steady states, by driving the system out of equilibrium, by extending them to more general noisy spin chains, by adding temperature effects, etc. We hope to report on these questions in a (possibly near) future.

## Acknowledgements

This work was in part supported by the CNRS and by the ANR project with contract number ANR-14-CE25-0003. The authors thank Marko Medenjak for useful discussions.

## A  Stationarity and correlations at order 3

Moments of higher orders can be computed following the method we used for the moments of first and second order. For third order moments we need to compute $\mathbb{E}_\infty(G_{ij}G_{kl}G_{mn})$. Again, because of the diffusive nature of the dynamics, we can regroup the terms that will have a non zero value in the stationary state in different groups, all elements within a given group having the same value. For the third order moments, the different groups are ; $\left[\,\bigcirc\,\right] := \mathbb{E}_\infty(G_{ij}G_{jk}G_{ki})$, $\forall(i\neq j\neq k)$ $\left[\,\bigcirc\bigcirc\,\right] := \mathbb{E}_\infty(G_{ii}G_{jk}G_{kj})$, $\forall(i\neq j\neq k)$, $\left[\,\bigcirc\!\bigcirc\,\right] := \mathbb{E}_\infty(G_{ij}G_{ji}G_{ii})$, $\forall(i\neq j)$, $\left[\,\bigcirc\bigcirc\bigcirc\,\right] := \mathbb{E}_\infty(G_{ii}G_{jj}G_{kk})$, $\forall(i\neq j\neq k)$, $\left[\,\bigcirc\!\bigcirc\!\bigcirc\,\right] := \mathbb{E}_\infty(G_{ii}^2 G_{jj})$, $\forall i\neq j$, $\left[\,\bigcirc\,\right] := \mathbb{E}_\infty(G_{ii}^3)$, $\forall i$. The dynamical equation of $\mathbb{E}_\infty(G_{ij}G_{kl}G_{mn})$ evaluated in the steady states imposes three relations between these quantities :

$$
\begin{aligned}
0 &= \left[\,\bigcirc\bigcirc\,\right] - \left[\,\bigcirc\!\bigcirc\,\right] + \left[\,\bigcirc\,\right], \\
0 &= \left[\,\bigcirc\!\bigcirc\!\bigcirc\,\right] - \left[\,\bigcirc\,\right] + 2\left[\,\bigcirc\!\bigcirc\,\right], \\
0 &= \left[\,\bigcirc\!\bigcirc\!\bigcirc\,\right] - \left[\,\bigcirc\bigcirc\bigcirc\,\right] - \left[\,\bigcirc\bigcirc\,\right].
\end{aligned}
\tag{18}
$$

We again have conserved quantities $N_3$, $N_2 N_1$, $N_1^3$:

$$
\begin{aligned}
N_1^3 &= L(L-1)(L-2)\left[\,\bigcirc\bigcirc\bigcirc\,\right] + 3L(L-1)\left[\,\bigcirc\!\bigcirc\!\bigcirc\,\right] + L\left[\,\bigcirc\,\right], \\
N_2 N_1 &= L(L-1)(L-2)\left[\,\bigcirc\bigcirc\,\right] + L(L-1)\left[\,\bigcirc\!\bigcirc\!\bigcirc\,\right] + L\left[\,\bigcirc\,\right] + 2L(L-1)\left[\,\bigcirc\!\bigcirc\,\right], \\
N_1^3 &= L(L-1)(L-2)\left[\,\bigcirc\,\right] + 3L(L-1)\left[\,\bigcirc\!\bigcirc\,\right] + L\left[\,\bigcirc\,\right].
\end{aligned}
$$

Together with the three previous equations they form a system of six independent equations that fix the value of our six unknowns;

$$\left[\vcenter{\hbox{⬡}}\right] = \frac{2N_3 + N_1^3 + 3N_2 N_1}{L(L+1)(L+2)},$$

$$\left[\vcenter{\hbox{◯◯◯}}\right] = \frac{-2N_3 + (L+1)N_1^3 + (L-1)N_1 N_2}{(L-1)L(L+1)(L+2)},$$

$$\left[\vcenter{\hbox{◯◯◯}}\right] = \frac{4N_3 + \left(L^2-2\right)N_1^3 - 3LN_1 N_2}{(L-2)(L-1)L(L+1)(L+2)},$$

$$\left[\vcenter{\hbox{◯}}\right] = \frac{LN_3 + (L-1)N_1 N_2 - N_1^3}{(L-1)L(L+1)(L+2)},$$

$$\left[\vcenter{\hbox{◯◯}}\right] = \frac{\left(L^2+2\right)N_1 N_2 - L\left(2N_3 + N_1^3\right)}{(L-2)(L-1)L(L+1)(L+2)},$$

$$\left[\vcenter{\hbox{◯}}\right] = \frac{N_3 L^2 - 3LN_1 N_2 + 2N_1^3}{(L-2)(L-1)L(L+1)(L+2)}.$$

We now calculate explicitly $\mathbb{E}_\infty[(\mathrm{tr}\,AG)^2]$ and $\mathbb{E}_\infty[\mathrm{tr}\,AG)^3]$ and show that only the term invariant under $U(L)$ conjugation remain :

$$\mathbb{E}_\infty[(\mathrm{tr}\,AG)^2] = \sum_{i,j,k,l} A_{i;j} G_{j;i} A_{k;l} G_{l;k}$$

$$= [\vcenter{\hbox{◯◯}}]\sum_i A_{i;i}^2 + [\vcenter{\hbox{◯◯}}]\sum_{i\neq j} A_{i;i} A_{j;j} + [\vcenter{\hbox{◯}}]\sum_{i\neq j} A_{i;j} A_{j;i}$$

$$= [\vcenter{\hbox{◯◯}}](\mathrm{tr}\,A_d^2) + [\vcenter{\hbox{◯◯}}]((\mathrm{tr}\,A)^2 - \mathrm{tr}\,A_d^2) + [\vcenter{\hbox{◯}}](\mathrm{tr}\,A^2 - (\mathrm{tr}\,A_d^2))$$

$$= [\vcenter{\hbox{◯◯}}](\mathrm{tr}\,A)^2 + [\vcenter{\hbox{◯}}](\mathrm{tr}\,A^2) + ([\vcenter{\hbox{◯◯}}] - [\vcenter{\hbox{◯◯}}] - [\vcenter{\hbox{◯}}])(\mathrm{tr}\,A_d^2),$$

where $A_d$ is defined by $A_{d\,i;j} = \delta_{i;j} A_{i;j}$. Recall that the condition we had on the different stationary values was $[\vcenter{\hbox{◯◯}}] = [\vcenter{\hbox{◯◯}}] + [\vcenter{\hbox{◯}}]$ so :

$$\mathbb{E}_\infty[(\mathrm{tr}\,AG)^2] = \left[\vcenter{\hbox{◯◯}}\right](\mathrm{tr}\,A)^2 + \left[\vcenter{\hbox{◯}}\right](\mathrm{tr}\,A^2),$$

and indeed only depends on the invariants. In the same manner one shows :

$$\mathbb{E}_\infty[(\mathrm{tr}\,AG)^3] = \left[\vcenter{\hbox{⬡}}\right]\mathrm{tr}\,A_d^3 + 3\left[\vcenter{\hbox{◯◯◯}}\right](\mathrm{tr}\,A_d^2\,\mathrm{tr}\,A - \mathrm{tr}\,A_d^3)$$

$$+ \left[\vcenter{\hbox{◯◯◯}}\right]((\mathrm{tr}\,A)^3 + 2\,\mathrm{tr}\,A_d^3 - 3\,\mathrm{tr}\,A_d^2\,\mathrm{tr}\,A)$$

$$+ 3\left[\vcenter{\hbox{◯◯}}\right](\mathrm{tr}\,A^2\,\mathrm{tr}\,A - 2\,\mathrm{tr}\,A^2 A_d - \mathrm{tr}\,A_d^2\,\mathrm{tr}\,A + 2\,\mathrm{tr}\,A_d^3)$$

$$+ 6\left[\vcenter{\hbox{◯◯}}\right](\mathrm{tr}\,A^2 A_d - \mathrm{tr}\,A_d^3)$$

$$+ 2\left[\vcenter{\hbox{◯}}\right](\mathrm{tr}\,A^3 - 3\,\mathrm{tr}\,A^2 A_d + 2\,\mathrm{tr}\,A_d^3)$$

$$= \left[\vcenter{\hbox{◯◯◯}}\right](\mathrm{tr}\,A)^3 + 3\left[\vcenter{\hbox{◯◯}}\right]\mathrm{tr}\,A^2\,\mathrm{tr}\,A + 2\left[\vcenter{\hbox{◯}}\right]\mathrm{tr}\,A^3$$

$$+ ([\vcenter{\hbox{⬡}}] - 3[\vcenter{\hbox{◯◯◯}}] + 2[\vcenter{\hbox{◯◯◯}}] + 6[\vcenter{\hbox{◯◯}}] - 6[\vcenter{\hbox{◯◯}}] + 4[\vcenter{\hbox{◯}}])\,\mathrm{tr}\,A_d^3$$

$$+ (3[\vcenter{\hbox{◯◯◯}}] - 3[\vcenter{\hbox{◯◯◯}}] - 3[\vcenter{\hbox{◯◯}}])\,\mathrm{tr}\,A_d^2\,\mathrm{tr}\,A$$

$$+ (-6[\vcenter{\hbox{◯◯}}] + 6[\vcenter{\hbox{◯◯}}] - 6[\vcenter{\hbox{◯}}])\,\mathrm{tr}\,A^2 A_d$$

$$= \left[\vcenter{\hbox{◯◯◯}}\right](\mathrm{tr}\,A)^3 + 3\left[\vcenter{\hbox{◯◯}}\right]\mathrm{tr}\,A^2\,\mathrm{tr}\,A + 2\left[\vcenter{\hbox{◯}}\right]\mathrm{tr}\,A^3.$$

To write the last line we used the stationary condition (18). Once again, only the invariant terms contribute.

# B  Proof of Proposition 1

We here present a proof of Proposition 1.

(i) The crucial observation is that the matrices $E_{j+1;j}$ and $E_{j;j+1}$, $j \in [1, L]$ which appear as the coefficients of Brownian motions in the definition of the process $V_t$, generate the Lie algebra $\mathfrak{sl}(L, \mathbb{C})$, so that the matrices $E_{j+1;j} + E_{j;j+1}$ and $i(E_{j+1;j} - E_{j;j+1})$, $j = 1, L$ generate the Lie algebra $\mathfrak{su}(L)$. Rewritten in terms of these generators, the coefficients will be independent real Brownian motions.

By Hörmander's theorem [32], we may conclude that the transition Kernel $K_t(v, B)$, which gives the probability that starting at $v$ at time $0$ $V_t$ will be in a Borel subset $B \subset SU(L)$, has a density with respect to the normalized Haar measure on $SU(L)$: $K_t(v, B) = \int_B k_t(v, v') d\eta(v')$ with $k_t(v, v')$ a continuous strictly positive function of $v' \in SU(L)$.

By homogeneity, $K_t(v, B) = K_t(1, v^{-1}B)$ and $k_t(v, v') = k_t(1, v^{-1}v')$. As $SU(L)$ is compact,

$$\inf_{v'} k_t(v, v') = \inf_{v'} k_t(1, v^{-1}v') = \inf_{v'} k_t(1, v') =: \varepsilon_t > 0$$

for every $t > 0$. Let $\nu$ be a finite (not necessarily positive) measure on $SU(L)$. Let us recall that by the Hahn-Jordan decomposition theorem there is a unique decomposition $\nu = \nu^+ - \nu^-$ where $\nu^{\pm}$ are finite positive measures, and that moreover there is a partition of $SU(L)$, $SU(L) = A^+ \cup A^-$, such that for every Borel subset of $SU(L)$ $\nu^{\pm}(B) = \pm \nu(B \cap A^{\pm})$. As usual, we define $|\nu| := \nu^+ + \nu^-$, the total variation measure of $\nu$ and denote by $||\nu||$ the total variation norm of $\nu$, i.e. $||\nu|| := \int_{SU(L)} d|\nu|$. We define another measure $\nu_t$, $t \geq 0$ by $\nu_t(B) := \int_{SU(L)} d\nu(v) K_t(v, B)$ for every Borel subset. Now if $\int_{SU(L)} d\nu(v) = 0$ then $\nu_t(B) = \int_{SU(L)} d\nu(v) (K_t(v, B) - \varepsilon_t \eta(B))$. Choosing $A^+, A^-$ that implement the Hahn-Jordan decomposition of $\nu_t$ we get

$$0 \leq \nu_t(A^+) = \int_{SU(L)} d\nu(v) \left(K_t(v, A^+) - \varepsilon_t \eta(A^+)\right) \leq \int_{SU(L)} d|\nu|(v) \left(K_t(v, A^+) - \varepsilon_t \eta(A^+)\right),$$

where we have used that $K_t(v, B) - \varepsilon_t \eta(B) \geq 0$ for every Borel set $B$, and analogously

$$0 \leq -\nu_t(A^-) = \int_{SU(L)} -d\nu(v) \left(K_t(v, A^-) - \varepsilon_t \eta(A^-)\right) \leq \int_{SU(L)} d|\nu|(v) \left(K_t(v, A^-) - \varepsilon_t \eta(A^-)\right).$$

Summing the two yields

$$||\nu_t|| \leq \int_{SU(L)} d|\nu|(v) \left(K_t(v, A^+ \cup A^-) - \varepsilon_t \eta(, A^+ \cup A^-)\right) = \int_{SU(L)} d|\nu|(v)(1 - \varepsilon_t) = ||\nu||(1 - \varepsilon_t)$$

for $t \geq 0$. We have proven that if $\int_{SU(L)} d\nu(v) = 0$ then $||\nu_t||$ decreases with $t$, and then by iteration that it decreases exponentially (for instance taking $\tau$ small and $t \in [n\tau, (n+1)\tau[$ we get

$$||\nu_t|| \leq ||\nu_{n\tau}|| \leq ||\nu||(1 - \varepsilon_\tau)^n \leq ||\nu||(1 - \varepsilon_\tau)^{t/\tau - 1}.$$

Now note that as time evolution acts by unitary multiplication on $V$, the Haar measure $d\eta()$ is obviously stationary. If $\mu$ is any initial probability distribution, $\nu := \mu - \eta$ has zero average, and thus $||\mu_t - \eta|| = ||\mu_t - \eta_t||$ decreases exponentially at large $t$, i.e. $\mu_t \to \eta$ in total variation norm. This implies also that $\eta$ is the only stationary measure for the stochastic flow $V_t$. This proves the statement concerning $V_t$.

The statement concerning $G_t$ follows immediately.

(ii) The invariance of $\mathbb{E}_\infty$ with respect to the flow generated by $dh_t$ means that its generating function satisfies

$$Z(A) = \mathbb{E}\left[Z\left(e^{idh_t} A e^{-idh_t}\right)\right], \tag{19}$$

where the expectation $\mathbb{E}$ is with respect to the Brownian increments $dW_t^j$ and $d\overline{W}_t^j$. Recall that these increments are Gaussian variables normalized according to $dW_t^j \, d\overline{W}_t^k = \delta^{j;k} dt$. Let us introduce some notations to express conveniently the consequences of this relation. For any Hermitian $X \in su(L)$, let $\mathcal{L}[X]$ be the vector field, linear in $X$, acting on function $F$ of $A$ via

$$\mathcal{L}[X]F(A) = \frac{d}{ds}F(e^{isX}Ae^{-isX})|_{s=0} = i\sum_{kl}[X,A]_{kl}\frac{\partial F(A)}{\partial A_{kl}}.$$

Recall that we choose $A$ to be Hermitian (or anti-Hermitian), so that the conjugation $A \to e^{isX}Ae^{-isX}$ preserves this property. These operators are anti-Hermitian with respect to the $\mathbb{L}^2$ scalar product, $(G, F) = \int dA \, \overline{G(A)} F(A)$. They form a representation of the Lie algebra $su(L) : [\mathcal{L}[X], \mathcal{L}[Y]] = \mathcal{L}[[X, Y]]$. We extend the definition of $\mathcal{L}$ to complex matrices by linearity, and let $\mathcal{L}_j^+ = \mathcal{L}[E_{j;j+1}]$ and $\mathcal{L}_j^- = \mathcal{L}[E_{j+1;j}]$. Now, expanding the expectation value (19) using the fact the Brownian increments are Gaussian variables with covariance $dt$, or using the Itô rules, yields

$$\frac{D}{2}\sum_j \left(\mathcal{L}_j^+ \mathcal{L}_j^- + \mathcal{L}_j^- \mathcal{L}_j^+\right) Z(A) = 0.$$

The differential operators $\mathcal{L}_j^+$ and $\mathcal{L}_j^-$ are hermitian conjugated with respect to the $\mathbb{L}^2$ scalar product, and the operators $\mathcal{L}_j^+ \mathcal{L}_j^-$ and $\mathcal{L}_j^- \mathcal{L}_j^+$ are all negative. Hence demanding the above equation be satisfied imposes

$$\mathcal{L}_j^\pm Z(A) = 0, \quad \forall j.$$

Since $E_{j;j+1}$ and $E_{j+1;j}$ form a system of simple root generators for the Lie algebra $su(L)$, the above equation implies that

$$\mathcal{L}[X]Z(A) = 0, \quad \forall X \in su(L).$$

Hence, $Z(A)$ is $SU(L)$ invariant and it is also $U(L)$ invariant because the extra $U(1)$ is central.

(iii) Using the $U(L)$ invariance and the fact that the spectrum of $G_t$ is preserved by the flow, we have

$$
\begin{aligned}
Z(A) &= \int d\eta(V) \quad Z(VAV^\dagger) = \int d\eta(V) \, \mathbb{E}_\infty\left[e^{\mathrm{tr}\,VAV^\dagger G}\right] \\
&= \mathbb{E}_\infty\left[\int d\eta(V)e^{\mathrm{tr}\,AV^\dagger GV}\right] = \mathbb{E}_\infty\left[\int d\eta(V)e^{\mathrm{tr}\,AV^\dagger G_0 V}\right] = \int d\eta(V) \quad e^{\mathrm{tr}\,AV^\dagger G_0 V}.
\end{aligned}
$$

In the first line we use the $U(L)$ invariance of $Z(A)$ and its definition as the generating function for $\mathbb{E}_\infty$. In the second line, we first permute the two integrations, with respect to the Haar measure and to $\mathbb{E}_\infty$, and second we use the fact that $\int d\eta(V)e^{\mathrm{tr}\,AV^\dagger GV}$ is a function of the spectrum of $G$ only (because it is invariant under conjugation of $G$ by a $U(L)$ matrix thanks to the invariance of the Haar measure). The last step in the third line consists in using the key property that the spectrum of $G$ is conserved by the flow and thus non random, so that $\int d\eta(V)e^{\mathrm{tr}\,AV^\dagger G_0 V}$ can be pulled out form the expectation with respect to $\mathbb{E}_\infty$.

## C  Invariant theory and expectations

Having recognized that any stationary measure for the time evolution of $G$ has to be $U(L)$ invariant (in fact $SU(L)$ invariant, but this makes no difference for the adjoint action), if $G_0$ is deterministic, or more generally if it it sampled within a set of unitarily equivalent matrices,

the stationary measure is exactly the one induced by the Haar measure on $U(L)$ via the adjoint action of $U(L)$ on $G_0$. This is the situation we concentrate on in this section.

So we forget about the time evolution, and simply ask: if $G$ is an $L \times L$ matrix, and if ${}^V G := V G V^{-1}$ i.e.

$$\left({}^V G\right)_{ij} := \sum_{k,l=1}^{L} V_{ik} G_{kl} V_{lj}^{-1}$$

is the action of the unitary matrix $V \in U(L)$ on the matrix $G$, what are the properties of the averages of functions of ${}^V G$ with respect to the normalized Haar measure $d\eta(V)$ on $U(L)$?

## C.1 Generalities

Recall that if $f$ is a (gentle) function from the set of $L \times L$ matrices to an affine space, we have defined

$$[f(G)] := \int_{U(L)} d\eta(V) f({}^V G).$$

We shall be in particular interested in the case when $f(G) := G_{i_1 j_1} \cdots G_{i_n j_n}$ for arbitrary $n = 1, 2, \cdots$ and $i_1, j_1, \cdots, i_n, j_n \in [1, L]$ or equivalently with an index-free notation $f(G) := G^{\otimes n}$. Note that $\left(G^{\otimes n}\right)_{i_1 j_1, \cdots, i_n j_n} = G_{i_1 j_1} \cdots G_{i_n j_n}$ and that $\left[\left(G^{\otimes n}\right)_{i_1 j_1, \cdots, i_n j_n}\right] = \left[G^{\otimes n}\right]_{i_1 j_1, \cdots, i_n j_n}$.

We can go a bit further in abstraction by noting that the matrix $G$ can be seen as a member of $\mathrm{End}(E) \cong E^* \otimes E$ where $E$ is the fundamental representation of $U(L)$ – $V$ acts as $(V.x)_i := \sum_j V_{ij} x_j$ – and $E^*$ its dual – $V$ acts as $(V.x^*)_i := \sum_j V_{ji}^{-1} x_j^*$. Observing that ${}^t V^{-1} \otimes V$ belongs to $\mathrm{End}(E^* \otimes E)$, so that $\left({}^t V^{-1} \otimes V\right)(G) := V G V^{-1}$ is naturally defined, $\left[G^{\otimes n}\right]$ can be retrieved by contracting appropriately the indices of $\int_{U(L)} d\eta(V) \left({}^t V^{-1} \otimes V\right)^{\otimes n}$ with those of $G^{\otimes n}$. Arrived at this stage, we may as well work not only with $G^{\otimes n}$ but with general elements $O \in (E^* \otimes E)^{\otimes n}$, that is, with general tensors. There is a natural action of $U(L)$ on $(E^* \otimes E)^{\otimes n}$, which in components reads

$$\left({}^V O\right)_{i_1 j_1, \cdots, i_n j_n} := \sum_{k_1, \cdots, k_n, l_1, \cdots, l_n}^{V} O_{k_1 l_1, \cdots, k_n l_n} V_{i_1 k_1} \cdots V_{i_n k_n} V_{l_1 j_1}^{-1} \cdots V_{l_n j_n}^{-1}.$$

For later use we introduce a transposition ${}^t$ mapping $(E^* \otimes E)^{\otimes n}$ to itself and reading in components

$$\left({}^t O\right)_{i_1 j_1, \cdots, i_n j_n} := O_{j_1 i_1, \cdots, j_n i_n},$$

and the corresponding adjoint $O^\dagger := {}^t \overline{O}$ where the bar is complex conjugation, a trace $\mathrm{Tr}$ (a linear form on mapping $(E^* \otimes E)^{\otimes n}$, we reserve the notation $\mathrm{tr}$ for the case $n = 1$) which in components reads

$$\mathrm{Tr}\, O := \sum_{i_1, \cdots, i_n} O_{i_1 i_1, \cdots, i_n i_n},$$

and a natural product $ON \in (E^* \otimes E)^{\otimes n}$ for $O, N \in (E^* \otimes E)^{\otimes n}$ which in components reads

$$(ON)_{i_1 j_1, \cdots, i_n j_n} := \sum_{k_1, \cdots, k_n} O_{i_1 k_1, \cdots, i_n k_n} N_{k_1 j_1, \cdots, k_n j_n}.$$

All these objects are the natural generalization of their ancestor for $n = 1$ and share most of its properties. For instance, it is plain that $\mathrm{Tr}\, {}^V O = \mathrm{Tr}\, O$ for every $V \in U(L)$, $\mathrm{Tr}\, {}^t O = \mathrm{Tr}\, O$, ${}^V(ON) = {}^V O\, {}^V N$ and $\mathrm{Tr}\, ON = \mathrm{Tr}\, NO$. Moreover, the sesquilinear form $(O, N) \mapsto \mathrm{Tr}\, O^\dagger N$ (which reduces to $\mathrm{Tr}\, {}^t ON$ for real tensors) is positive definite.

We extend the definition of $[\cdots]$ to $(E^* \otimes E)^{\otimes n}$

$$[O] := \int_{U(L)} d\eta(V)\,{}^V O,$$

which yields an endomorphism of $(E^* \otimes E)^{\otimes n}$ and we observe that by the left invariance of the Haar measure $[O]$ is invariant for the action of $U(L)$, i.e. that ${}^V[O] = [O]$ for every $V \in U(L)$.

We now come to the crux of the matter. The identity $V^{-1}V = \mathrm{Id}$ can be reinterpreted as that ${}^V\mathrm{Id} = \mathrm{Id}$, i.e. contracting ${}^tV^{-1}$ and $V$ in ${}^tV^{-1} \otimes V$ using $\mathrm{Id}$ yields $\mathrm{Id}$: $\mathrm{Id}$ is an invariant for the action of $U(L)$ on $\mathrm{End}(E)$. It is well-known, and easy to prove, that this is the only invariant up to normalization. This has a nice generalization to

$$\left({}^tV^{-1} \otimes V\right)^{\otimes n} = {}^tV^{-1} \otimes V \otimes {}^tV^{-1} \otimes V \otimes \cdots \otimes {}^tV^{-1} \otimes V :$$

if one contracts each of the $n$ ${}^tV^{-1}$s in the product with a $V$ (the $V$s will be different for different ${}^tV^{-1}$s of course, there are only enough indices to contract once) one obtains an invariant. We interpret the contraction pattern in a natural way as a permutation $\sigma \in \mathfrak{S}_n$ of $[1, n]$ such that (reading from left to right) the $i^{th}$ factor ${}^tV^{-1}$ is contracted with the $\sigma(i)^{th}$ factor $V$, so that the resulting invariant $I^\sigma$ reads in components

$$I^\sigma_{i_1 j_1, \cdots, i_n j_n} := \delta_{i_{\sigma(1)} j_1} \cdots \delta_{i_{\sigma(n)} j_n}.$$

Note that

$$\mathrm{Tr}\, I^\sigma = L^{c(\sigma)},$$

where $c(\sigma)$ is the number of cycles of the permutation $\sigma$. One checks readily that $I^{\sigma^{-1}} = {}^t(I^\sigma)$ for $\sigma \in \mathfrak{S}_n$ and $I^\sigma I^\tau = I^{\sigma\tau}$ for $\sigma, \tau \in \mathfrak{S}_n$.

It turns out (see e.g. chapter 5 in [38]) that for general $n$ (this is quite a bit deeper that the special case $n = 1$) the space of invariants is spanned by the $I^\sigma$s when $\sigma$ ranges over $\mathfrak{S}_n$.

If $n \leq L$ they are linearly independent: if $i_1, \cdots, i_n$ are all distinct, $I^\sigma_{i_1 i_1, \cdots, i_n i_n}$ vanishes for every permutation except the identity, so that the invariant corresponding to the trivial permutation is linearly independent from the other invariants, and from $I^\sigma I^\tau = I^{\sigma\tau}$ one infers the full linear independence.

If $n > L$ this is not true anymore. We do not reprove this and content to observe that if $n$ is so large that $n! > L^{2n}$ there are more $I^\sigma$s than the dimension of $(E^* \otimes E)^{\otimes n}$ and they cannot be linearly independent.

Henceforth we assume that $n \leq L$. The linear independence of the $I^\sigma$s has the following two consequences. First, the matrix $C$ with rows and columns indexed by $\mathfrak{S}_n$ and matrix elements $L^{c(\sigma\tau^{-1})}$ is positive definite, because

$$C_{\sigma,\tau} := L^{c(\sigma\tau^{-1})} = \mathrm{Tr}\, I^\sigma\, {}^t I^\tau.$$

Second every invariant tensor is an unique linear combination of the $I^\sigma$s. We denote by $\left({}^tV^{-1} \otimes V\right)^{\otimes n, \mathrm{inv}}$ the space of invariant tensors. Now $[O]$ is invariant for every $O \in (E^* \otimes E)^{\otimes n}$, and we infer the existence and uniqueness of linear forms $\ell_\sigma$ on $(E^* \otimes E)^{\otimes n}$ such that

$$[O] = \sum_{\sigma \in \mathfrak{S}_n} \ell_\sigma(O) I^\sigma.$$

Let us pause for a moment to stress one feature this formula makes obvious: the correlation functions are "topological". In the formulation of the model, the sites $i = 1, \cdots, L$ are arranged around a ring, and the form of the interactions gives a physical meaning to the notion that site $i$ is connected to sites $i \pm 1$ i.e. that those sites are neighbors. However, the $U(L)$ Haar measure

does not care about neighbors anymore: after all, the $L \times L$ permutation matrices belong to $U(L)$ so we expect that if $\pi$ is any permutation of $[1, L]$

$$[O]_{i_1 j_1, i_2 j_2 \cdots, i_n j_n} = [O]_{\pi(i_1)\pi(j_1),\pi(i_2)\pi(j_2)\cdots,\pi(i_n)\pi(j_n)},$$

which is clearly true from the explicit form of the matrix elements of each $I^\sigma$.

The $\ell_\sigma$s could in principle be computed without any recourse to integration by an usual trick: for each $\tau \in \mathfrak{S}_n$ we have

$$
\begin{aligned}
\mathrm{Tr}\left[OI^{\tau^{-1}}\right] &= \mathrm{Tr}\int_{U(L)} d\eta(V)\,{}^V(OI^{\tau^{-1}}) = \int_{U(L)} d\eta(V)\,\mathrm{Tr}\,{}^V(OI^{\tau^{-1}}) \\
&= \int_{U(L)} d\eta(V)\,\mathrm{Tr}\,OI^{\tau^{-1}} = \mathrm{Tr}\,OI^{\tau^{-1}},
\end{aligned}
$$

while

$$\mathrm{Tr}\sum_{\sigma\in\mathfrak{S}_n}\ell_\sigma(O)I^\sigma I^{\tau^{-1}} = \sum_{\sigma\in\mathfrak{S}_n}\ell_\sigma(O)L^{c(\sigma\tau^{-1})}.$$

Comparison yields

$$\sum_{\sigma\in\mathfrak{S}_n}\ell_\sigma(O)L^{c(\sigma\tau^{-1})} = \mathrm{Tr}\,OI^{\tau^{-1}}.$$

As noted above, the matrix $C_{\sigma,\tau} := L^{c(\sigma\tau^{-1})}$ is positive definite, hence invertible, and

$$\ell_\sigma(O) = \sum_{v\in\mathfrak{S}_n}\mathrm{Tr}\,OI^{v^{-1}}C_{v,\sigma}^{-1}.$$

Thus in principle the task of computing $U(L)$ averages is reduced to the inversion of the matrix $C$. The last formula has two immediate consequences. First, if $O$ is orthogonal (with respect to the trace form) to $\left({}^tV^{-1}\otimes V\right)^{\otimes n,\mathrm{inv}}$ then $\ell_\sigma(O) = 0$ for every $\sigma$. Second, taking $O$ to be some $I^\tau$ yields

$$\ell_\sigma(I^\tau) = \sum_{v\in\mathfrak{S}_n}\mathrm{Tr}\,I^\tau I^{v^{-1}}C_{v,\sigma}^{-1} = \sum_{v\in\mathfrak{S}_n}C_{\tau,v}C_{v,\sigma}^{-1} = \delta_\sigma^\tau.$$

Thus, restricted to $\left({}^tV^{-1}\otimes V\right)^{\otimes n,\mathrm{inv}}$ the linear forms $\ell_\sigma$ build the dual basis of the basis of invariants $I^\sigma$.

The explicit inverse of $C$ is not so easy to write down in general, and the size of $C$, $n!$, makes computations prohibitive even for moderate $n$s. However in the end, our interest is in tensors $O$ of the form $O = G^{\otimes n}$, and the linear space they span is the space of symmetric tensors $\left({}^tV^{-1}\otimes V\right)^{\otimes n}_{sym}$. If $O \in \left({}^tV^{-1}\otimes V\right)^{\otimes n}$ and $\sigma\in\mathfrak{S}_n$ we define $\sigma\cdot O$ by

$$(\sigma\cdot O)_{i_1 j_1,\cdots,i_n j_n} := O_{i_{\sigma(1)}j_{\sigma(1)},\cdots,i_{\sigma(n)}j_{\sigma(n)}},$$

which is a left action, i.e. $\tau\cdot(\sigma\cdot O) = (\tau\sigma)\cdot O$. Symmetric tensors are those $O$s such that $\sigma\cdot O = O$ for every $\sigma\in\mathfrak{S}_n$. A simple computation shows that $\sigma\cdot I^\tau = I^{\sigma\tau\sigma^{-1}}$, $\sigma\cdot({}^VO) = {}^V(\sigma\cdot O)$ and then $[\sigma\cdot O] = \sigma\cdot[O]$, so we infer that $\ell_\tau(\sigma\cdot O) = \ell_{\sigma^{-1}\tau\sigma}(O)$. In particular, restricted to $\left({}^tV^{-1}\otimes V\right)^{\otimes n}_{sym}$, $\ell_\sigma$ depends only on the conjugacy class of $\sigma$ in $\mathfrak{S}_n$. We define the cycle spectrum of a permutation $\sigma\in\mathfrak{S}_n$ as the collection $n_k = n_k(\sigma)$, $k \geq 1$ where $n_k$ is the number of cycles of length $k$ in the cycle decomposition of $\sigma$. The $n_k$s satisfy $\sum_k kn_k = n$. Two members in $\mathfrak{S}_n$ are conjugate if and only if they have the same cycle lengths spectrum, so conjugacy classes in $\mathfrak{S}_n$ are parametrized by integers sequences $n_k, k \geq 1$ such that $\sum_k kn_k = n$. These sequences also parameterize Young diagrams with $n$ boxes ($n_k$ is the number of rows of length $k$ in the

diagram) or (unordered) partitions of $n$. Thus in the case of symmetric tensors the complexity is reduced from $n!$ to $p(n)$, the number of partitions of $n$. Henceforth we denote by the same name a Young diagram and a conjugacy class, identifying a Young diagram with $n$ boxes with a subset of $\mathfrak{S}_n$.

Also, in the special case $O = G^{\otimes n}$ one checks that, if $\sigma \in \mathfrak{S}_n$ has cycle spectrum $n_k$, $k \geq 1$ then

$$\operatorname{Tr} G^{\otimes n} I^{\sigma^{-1}} = \prod_k \left(\operatorname{tr} G^k\right)^{n_k}.$$

If $\lambda$ is a Young diagram we set

$$I_\lambda := \sum_{\sigma \in \lambda} I^\sigma$$

and denote by $\ell^\lambda$ the restriction of $\ell_\sigma$ (for any $\sigma \in \lambda$) to the space of symmetric tensors. Thus if $O \in \left(\,^t V^{-1} \otimes V\right)^{\overset{sym}{\otimes n}}$ we have

$$[O] = \sum_\lambda \ell^\lambda(O) I_\lambda.$$

The $I_\lambda$s form a basis of symmetric invariant tensors, and the duality relation $\ell^\lambda(I_\mu) = \delta^\lambda_\mu$ holds at the level of conjugacy classes. Noting that in $\mathfrak{S}_n$ a permutation and its inverse are conjugate (they have obviously the same cycle lengths), we also obtain

$$\operatorname{Tr} O I_\mu = \sum_\lambda \ell^\lambda(O) C^{\lambda,\mu} \text{ where } C^{\lambda,\mu} := \sum_{\sigma \in \lambda, \tau \in \mu} C_{\sigma,\tau} = \sum_{\sigma \in \lambda, \tau \in \mu} L^{c(\sigma\tau^{-1})}.$$

Let us note that another symmetry condition, complete symmetry under permutations of the $i$s and/or the $j$s, which would be relevant in the study of the statistical properties of $\langle c_i \rangle$ and $\langle c_j^\dagger \rangle$, leads to the fact the $\ell_\sigma$s applied to symmetric objects are $\sigma$-independent and can thus be computed explicitly. This leads to a completely solvable case, with mostly pedagogical interest and we leave the details to the reader.

## C.2 Application to the one-site statistics of the fermion number

As noticed above, there is no easy closed form for the $\ell_\sigma$s. However, they satisfy certain sum rules. We give two of them, and then use the second one to give an explicit form for the averages of $G_{ii}^n$ for arbitrary $n$, i.e. moments of the particle number at site $i$.

We start with two counting formulæ:

$$\sum_{\sigma \in \mathfrak{S}_n} L^{c(\sigma)} \varepsilon(\sigma) = L(L-1)\cdots(L-n+1) \qquad \sum_{\sigma \in \mathfrak{S}_n} L^{c(\sigma)} = L(L+1)\cdots(L+n-1),$$

were $\varepsilon(\sigma)$ is the signature of the permutation $\sigma$ i.e. $\varepsilon(\sigma) = -1$ if the cycle decomposition of $\sigma$ contains an odd number of cycles of even length, and $\varepsilon(\sigma) = 1$ otherwise.

Induction on $n$ gives an easy proof: the formulæ are obvious if $n = 1$. To work out the induction step, write down the cycle decomposition of a permutation in $\mathfrak{S}_{n+1}$ and remove $n+1$ to get a permutation in $\mathfrak{S}_n$. If $n+1$ was a cycle by itself, removing it diminishes the number of cycles by 1 but does not change the signature. If $n+1$ was in a cycle of length $\geq 2$, removing it changes the signature but not the number of cycles. Now from a permutation in $\mathfrak{S}_n$ written as a product of cycles, there is only 1 way to insert $n+1$ as a new cycle on its own, and $n$ ways to insert it within already existing cycles. Thus going from $n$ to $n+1$ yields a factor $L-n$ for the first sum, and $L+n$ for the second sum.

We return to the general formula

$$\sum_{\sigma \in \mathfrak{S}_n} \ell_\sigma(O) L^{c(\sigma\tau^{-1})} = \operatorname{Tr} O I^{\tau^{-1}}.$$

Multiply by $\varepsilon(\tau) = \varepsilon(\tau^{-1}) = \varepsilon(\sigma)\varepsilon(\sigma\tau^{-1})$. On the left-hand side, use $\sigma\tau^{-1}$ for fixed $\sigma$ and varying $\tau$ as a summation variable. On the right-hand side, use keep $\tau^{-1}$ as a summation variable. This yields

$$\sum_{\sigma\in\mathfrak{S}_n} \ell_\sigma(O)\varepsilon(\sigma)L(L-1)\cdots(L-n+1) = \sum_{\tau\in\mathfrak{S}_n} \mathrm{Tr}\, OI^\tau \varepsilon(\tau).$$

The same change of variable applied to the general formula without multiplying by the signature yields a second identity

$$\sum_{\sigma\in\mathfrak{S}_n} \ell_\sigma(O)L(L+1)\cdots(L+n-1) = \sum_{\tau\in\mathfrak{S}_n} \mathrm{Tr}\, OI^\tau.$$

It is this last identity that we are going to exploit. We observe that for $k \in [1, L]$ the matrix element

$$\left[ O_{ii\cdots ii} \right] = \sum_{\sigma\in\mathfrak{S}_n} \ell_\sigma(O)I^\sigma_{ii\cdots ii} = \sum_{\sigma\in\mathfrak{S}_n} \ell_\sigma(O),$$

because from the definition of $I^\sigma$ the matrix element $I^\sigma_{ii\cdots ii}$ (all indices are equal to $k$) equals 1 for every $\sigma \in \mathfrak{S}_n$. Using the second identity we obtain

$$\left[ O_{ii\cdots ii} \right] = \frac{(L-1)!}{(L+n-1)!} \sum_{\tau\in\mathfrak{S}_n} \mathrm{Tr}\, OI^\tau.$$

Specializing to $O = G^{\otimes n}$, we can use our observations on symmetric tensors to simplify this formula. Let us recall that the number of permutations with cycle spectrum $n_k, k \geq 1$ (with $\sum_k kn_k = n$) is

$$\frac{n!}{\prod_k n_k! k^{n_k}}.$$

The counting is elementary. For example, if $m_1, \cdots, m_j$ are positive integers such that $\sum_{i=1}^j m_i = n$ there are $\frac{n!}{\prod_i m_i!}$ ways to color $n$ objects with $j$ colors, with $m_i$ objects carrying color $i$ for $1 = 1, \cdots, j$. Our interest is when $\sum_k n_k = j$, yielding $\frac{n!}{\prod_k (k!)^{n_k}}$ colorings. But in contrast to the pure coloring problem:
– We have to order each packet of a given color into a cycle, for a packet of size $k$ there are $(k-1)!$ ways to do so.
– We care only about the packets, not about their precise color, so we have to divide by $\prod_k n_k!$. Finally

$$\frac{n!}{\prod_k (k!)^{n_k}} \times \prod_k (k-1)!^{n_k} / \prod_k n_k! = \frac{n!}{\prod_k n_k! k^{n_k}},$$

as announced.

Recall that the diagonal elements of $G$ have a clear physical meaning: $G_{ii} = \langle c_i^\dagger c_i \rangle$, the (quantum) mean particle number at site $i$. Thus

$$\left[ \langle c_i^\dagger c_i \rangle^n \right] = \left[ G_{ii}^n \right] = \frac{n!(L-1)!}{(L+n-1)!} \sum_{n_k, \sum_{k\geq 1} kn_k = n} \prod_k \frac{1}{n_k!} \left( \frac{N_k}{k} \right)^{n_k},$$

proving the formula announced in the main text. It is strongly reminiscent of the combinatorics relating moments to cumulants, in that if $\mu_n, n \geq 0$ and $\gamma_k, k \geq 1$ are two sequences such that as formal series in $x$

$$\sum_{n\geq 0} \mu_n \frac{x^n}{n!} = e^{\sum_{k\geq 1} \gamma_k \frac{x^k}{k}},$$

it is readily checked that $\mu_0 = 1$ and that for $n \geq 1$

$$\mu_n = n! \sum_{n_k, \sum_{k \geq 1} k n_k = n} \prod_k \frac{1}{n_k!} \left( \frac{\gamma_k}{k} \right)^{n_k}.$$

Apart from the combinatorial factor $\frac{(L-1)!}{(L+n-1)!}$, the traces $N_k := \operatorname{tr} G^k$ are analogs of the cumulants of the distribution of $G_{ii} = \langle c_i^\dagger c_i \rangle$. The combinatorial factor is not totally innocent (i.e. cannot be reabsorbed by a trivial manipulation). But one checks that if $\mu_n := \left[ \langle c_i^\dagger c_i \rangle^n \right]$ then

$$\gamma_1 = \frac{N_1}{L} \quad \gamma_2 = \frac{N_2}{L(L+1)} - \frac{1}{L+1} \left( \frac{N_1}{L} \right)^2,$$

and so on.

In the large $L$ limit under the scaling $N_k \simeq L$ for $k = 0, 1, \cdots$ the asymptotics gives

$$\left[ \langle c_i^\dagger c_i \rangle^n \right] = \left( \frac{N_1}{L} \right)^n + \frac{n(n-1)}{2L} \left( \frac{N_1}{L} \right)^{n-2} \left( \frac{N_1}{L} - \left( \frac{N_2}{L} \right)^2 \right) + o(1/L),$$

whilde under the scaling $N_k \simeq L^k$; $k = 1, 2, \cdots$ all terms in the sum over partitions contribute to the dominant order, leading to

$$\left[ e^{x \langle c_i^\dagger c_i \rangle} \right] = e^{\sum_{k \geq 1} \frac{N_k x^k}{kL^k}} + o(1) = \frac{1}{\det(1 - xG/L)} + o(1).$$

## C.3 The covering rule decomposition

In this subsection we establish the labeled counterpart of Lemma 2, which reads: if $D$ is an admissible diagram then

$$[D] = \sum_{D' \text{ covering } D} [D']$$

(multiplicity is 1 for each covering diagram). The unlabeled version is an immediate consequence. The point is that the covering diagrams are always extremal diagrams. In the labeled category, they are associated to a $2n$-plets $(i_1, j_1, i_2, j_2 \cdots, i_n, j_n)$ with $i_1, \cdots, i_n$ must all distinct and a well-defined permutation $\sigma \in \mathfrak{S}_n$ such that $i_{\sigma(m)} = j_m$ for $m = 1, \cdots, n$. Then, for an extremal $2n$-plets $(i_1, j_1, i_2, j_2 \cdots, i_n, j_n)$

$$\left[ G^{\otimes n} \right]_{i_1 j_1, \cdots, i_n j_n} = \ell_\sigma(G^{\otimes n}).$$

But for an arbitrary $2n$-plets $(i_1, j_1, i_2, j_2 \cdots, i_n, j_n)$ we have the general formula

$$\left[ G^{\otimes n} \right] = \sum_{\sigma \in \mathfrak{S}_n} \ell_\sigma(O) I^\sigma.$$

From the very definition of a $*$-covering, $I^\sigma_{i_1 j_1, \cdots, i_n j_n} = 0$ unless $(i_1, j_1, i_2, j_2 \cdots, i_n, j_n)$ is admissible and has a $*$-covering of with associated permutation $\sigma$ in which case $I^\sigma_{i_1 j_1, \cdots, i_n j_n} = 1$. Putting the two equations together yields, for an arbitrary admissible $2n$-plets $(i_1, j_1, i_2, j_2 \cdots, i_n, j_n)$

$$\left[ G^{\otimes n} \right]_{i_1 j_1, \cdots, i_n j_n} = \sum_{(k_1, l_1, \cdots, k_n, l_n) \text{ covering } (i_1, j_1, \cdots, i_n, j_n)} \left[ G^{\otimes n} \right]_{k_1 l_1, \cdots, k_n l_n},$$

which translates immediately in terms of diagrams in the identity that was to be proven.

## D  Character expansion of $Z(A)$

Let us list the characters of the linear group up to $m(Y) = 3$.

| Y | $\chi_{Y(A)}$ | $d_Y$ | $\sigma_Y$ |
|---|---|---|---|
| $\square$ | $\mathrm{tr}A$ | $L$ | 1 |
| $\square\square$ | $\frac{1}{2}((\mathrm{tr}A)^2 + \mathrm{tr}A^2)$ | $\frac{1}{2}L(L+1)$ | 1 |
| $\square$ over $\square$ | $\frac{1}{2}((\mathrm{tr}A)^2 - \mathrm{tr}A^2)$ | $\frac{1}{2}L(L-1)$ | 1 |
| $\square\square\square$ | $\frac{1}{6}((\mathrm{tr}A)^3 + 2\,\mathrm{tr}A^3 + 3\,\mathrm{tr}A\,\mathrm{tr}A^2)$ | $\frac{1}{6}L(L+1)(L+2)$ | 1 |
| $\square\square$ / $\square$ | $\frac{1}{3}((\mathrm{tr}A)^3 - \mathrm{tr}A^3)$ | $\frac{1}{3}L(L+1)(L-1)$ | 2 |
| $\square$/$\square$/$\square$ | $\frac{1}{6}((\mathrm{tr}A)^3 + 2\,\mathrm{tr}A^3 - 3\,\mathrm{tr}A\,\mathrm{tr}A^2)$ | $\frac{1}{6}L(L-1)(L-2)$ | 1 |

We explicitly compute the term of degree three $Z(A)|_3$ in the character expansion of the Harish-Chandra-Itzykson-Zuber generating function and show that it matches the results obtained by explicit calculations of the coefficient by perturbative treatment.

$$
\begin{aligned}
Z(A)|_3 &= \frac{1}{6}\Big(\frac{6}{L(L+1)(L+2)}\,\square\square\square(A)\,\square\square\square(G_0) + \frac{3}{L(L+1)(L-1)}\,\square\square(A)\,\square\square(G_0) \\
&\quad + \frac{6}{L(L-1)(L-2)}\,\square(A)\,\square(G_0)\Big) \\
&= \frac{1}{6}\Big(\frac{1}{6L(L+1)(L+2)}((\mathrm{tr}A)^3 + 2\,\mathrm{tr}A^3 + 3\,\mathrm{tr}A\,\mathrm{tr}A^2)(N_1^3 + 2N_3 + 3N_1N_2) \\
&\quad + \frac{2}{3L(L+1)(L-1)}((\mathrm{tr}A)^3 - \mathrm{tr}A^3)(N_1^3 - N_3) \\
&\quad + \frac{1}{6L(L-1)(L-2)}((\mathrm{tr}A)^3 + 2\,\mathrm{tr}A^3 - 3\,\mathrm{tr}A\,\mathrm{tr}A^2)(N_1^3 + 2N_3 - 3N_1N_2))\Big) \\
&= \frac{1}{6}\Big((\mathrm{tr}A)^3\Big(\frac{(L^2-2)N_1^3 - 3LN_1N_2 + 4N_3}{(L-2)(L-1)L(L+1)(L+2)}\Big) \\
&\quad + 2(\mathrm{tr}A^3)\Big(\frac{2N_1^3 - 3LN_1N_2 + L^2N_3}{(L+2)(L-1)L(L+1)(L+2)}\Big) \\
&\quad + 3\,\mathrm{tr}A\,\mathrm{tr}A^2\Big(\frac{-LN_1^3 + (L^2+2)N_1N_2 + 2LN_3}{(L-2)(L-1)L(L+1)(L+2)}\Big)\Big) \\
&= \frac{1}{6}([\heartsuit\heartsuit\heartsuit](\mathrm{tr}A^3) + 3[\heartsuit\,\bigcirc]\,\mathrm{tr}A\,\mathrm{tr}A^2 + 2[\bigcirc]\,\mathrm{tr}A^3).
\end{aligned}
$$

## E  Large $L$ scaling limits

We turn to the possible scaling behaviors of $U(L)$ averages at large $L$. Let us start with an obvious bound. By definition, the matrix elements of $G$, $G_{ij} := \langle c_j^\dagger c_i \rangle$, have modulus $\leq 1$ by the Cauchy-Schwartz inequality. This implies immediately that $|\mathrm{tr}\,G^k| \leq L^k$ for $k = 1, 2, \cdots$. As these traces are the building blocks of correlation functions, any physical scaling limit at large $L$ must respect this constraint.

But the very notion of large $L$ scaling limit has to be taken with a grain of salt because it implies to work with a family of density matrices indexed by $L$, and this can only be based on physical assumptions.

To illustrate the point, let us observe that our model can be obtained via a Jordan-Wigner transformation from a spin model. The spin model looks like the fermionic one except that the Hilbert space is not a Fock space but a tensor product $\left(\mathbb{C}^2\right)^{\otimes L}$ where the fermionic operators commute at different sites (but have the usual anticommutation rules at a given site)[4]. Let $r$ be a one site density matrix possibly with non-trivial off-diagonal elements. Then a factorized density matrix $\rho = \rho(L) := r^{\otimes L}$ gives a natural candidate for which the system ought to have a large $L$ limit. An easy computation shows that if this density matrix is used for quantum averages then $\langle c_j^\dagger c_i \rangle$ (remember the fermionic operators commute at different sites in this discussion) is of the general form

$$\langle c_j^\dagger c_i \rangle = \alpha \delta_{ij} + \beta \text{ with } \alpha(1-\alpha) \geq \beta \geq 0.$$

Then $\operatorname{tr} G^k = \alpha^k (L-1) + (\alpha + \beta L)^k$. As $\beta$ turns out to be the modulus square of the off-diagonal element of the one site density matrix $r$ (in the basis where $c^\dagger c$ is diagonal), we infer that $\operatorname{tr} G^k \propto L$ if $r$ is diagonal, but $\operatorname{tr} G^k \propto L^k$ else.

Even if, for reasons recalled in the footnote, the above result does not translate immediately in Fock space (of course we could use $G$ to reverse-engineer an $M$ and the corresponding density matrix on Fock space but this is a bit artificial), we expect that those scaling behaviors are natural there as well. Other scaling behaviors are possible, but in the present study we have concentrated on the above two. The case when $\operatorname{tr} G^k \propto L$ for $k = 0, 1, \cdots$ means roughly that all eigenvalues of $G$ are of order 1. The case when $\operatorname{tr} G^k \propto L^k$ for $k = 1, 2, \cdots$, meaning roughly that all eigenvalues of $G$ are of order 1 but a finite number of them which are of order $L$.

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
