# Peer review of "Equilibrium Fluctuations in Maximally Noisy Extended Quantum Systems"

_SciPost Physics, doi:SciPost Phys. 6, 045 (2019)_

## Round 2 · Referee Report · Anonymous · 2018-12-27

Strengths

1- Exact results for a steady-state distribution of observables in a tight-binding model with stochastic hopping. Due to unitary invariance they can be expressed in terms of integrals known from RMT.

2- Could potentially be of relevance in a number of different contexts. Importantly, and the authors do comment (speculate) on that, is that the results could be valid more generally and not just for the specific model.

Weaknesses

Mostly minor presentational issues.

1- Perhaps more explicitly comment on related settings.

Report

As highlighted in the "Strengths" the paper presents very nice exact results. Besides providing an explicit result it also opens new questions, in particular about possible "universality" of fluctuations. I therefore recommend its publication.

Requested changes

Some optional minor comments (mostly regarding presentation) are:

1- In introduction, while the authors do connect with related settings, if possible, it would be nevertheless useful to be more explicit about the model studied. E.g., connection to random circuits (can one make an exact statement to what circuit eq.(1) corresponds to); that would be of interest as in (recent) quantum circuits community they make specific predictions about fluctuations (e.g., diffusive v. KPZ). To master equations (what kind of master eq. should one have in mind for eq.(2)).

On p.3, 3rd line, they likely mean "XX spin chain with dephasing", and not general XXZ (in order to get H_0 with hopping only like in eq.(2))?

2- 2nd paragraph on p.3: it could be miss-understood that only stochastic models give access to fluctuations, which is not the case. Large deviation formalism can be used also for master equations (and has been applied also for e.g. diffusive XX chain with dephasing, PRE 89, 042140 '14). It would be useful to mention some literature on that, review by H.Touchette, Phys. Rep. 478, 1 (2009) would be a good choice, especially because the authors anyway use/mention large deviation functions in the text.

3- It is worth considering to mention in the Introduction with respect to the H-C-I-Z integral also RMT or "integrals over unitary Haar measure" -- I think this will ring more bells (and attract readers) than just some technical name.

4- Using "perturbative" in e.g. title of Sec.2,3: A casual reader might misinterpret that e.g. Sec.2 are "just" perturbative results (in some small parameter), rather than **exact** results for low-order correlations. I understand that "perturbative" is meant with respect to expansion of the full distribution function, but using some other name would be better (similarly in the last paragraph on p.3 "...to first few orders." might seem to imply some perturbative expansion, while it means first few low-order correlations).

5- After eq.(8): readers familiar with RMT results, specifically averaging over unitary measure, will recognize those formulas as they appear in numerous contexts (thermalization, form factors, random channels...). Sequentially reading the paper it would help if already at that point the authors would reaffirm that this is so (even if they do explain it in details in the very next section).

6- 2nd sentence after eq.(8): explain what is meant by "topological nature".

7- Sentence before eq.(13): mention what q is.

8- Sec.4: can one compare the results obtained with some classical diffusive model (e.g. SSEP), what if anything is different?

9- Sec.5: considering that all the proofs and details are given in the appendices, I am wondering if also Sec.5 fits more in the appendix.

10- Sec.6 is very interesting. I see 1st results about independence on mu_j as showing that even with disorder a random hopping destroys any signs of localization. Next two paragraphs are intriguing. The authors argue that even with interactions the stationary distribution might stay the same. On the other hand, a (likely) scenario is also that the convergence radius goes to zero (with system size)? For exact results in the paper two conditions are crucial: (i) free model, (ii) unitary invariance. One can ask if both are necessary? At some point the authors do mention that the generators are ergodic (resulting in unitary invariance) only in the 1-particle sector, so a free nature of the model seems crucial. On the other hand, here in Sec.6, the argument seems to be that fluctuations will be the same provided generators L_j span the whole group (which means for almost any generic "noisy" evolution) -- that would mean a rather general universality of the results derived. An argument why this is perhaps too much to expect would be that fluctuations are on physical grounds connected to relaxation (if the regression theorem holds this is an exact statement) -- they can be different for slow vs. fast relaxation, while in the presented work there is no dependence on relaxation.

---

## Round 2 · Referee Report · Anonymous · 2019-2-12

Strengths

1. Contribution to study of noisy many body systems, of great current interest for theoretical (and possibly practical) reasons.

2. Introduces rigorous methods in harmony with a clear physical picture for the case of the stochastic XX model

3. Establishes a complete description of the steady state for this model, based on (i) Explicit computations for low cumulants of the single particle density matrix (ii) An exact description of the generating function of the same in terms of the Harish-Chandra-Itzykson-Zuber integral.

4. Discusses generalizations and potential generalizations to other models.

Weaknesses

1. It would be good to understand the results of the present work in terms of the authors' earlier work (Ref. 12) on the noisy Heisenberg chain. Since there is no free fermion picture here, it would be surprising if the present results apply, though that seems to be the implication after Eq. 1.

2. In the same way, the occupation number statistics at the end of Section 3 are presumably quite different from those in the stationary state of the symmetric exclusion process, where each site has an independent Bernoulli distribution.

Report

The study of the dynamics of noisy many body systems is relatively undeveloped, and the present work is a very useful step in the direction of understanding the steady state of a particular model. Exploiting the free fermion structure allows the steady state to be determined in great detail by a variety of methods. The statistics of the single particle density matrix are established in two physically relevant regimes.

Some generalizations to other models are discussed, including the more interesting case of a genuinely interacting system.

Requested changes

Can the meaning of "non random" be clarified in the description of the steady state?

Clarify relation to noisy Heisenberg spin chain. Ref. 12 doesn't actually discuss the relation mentioned after Eq. (1).

End of Section 1: "the rules of the diagrammatic..." What?

Small typo at the end of Section 4: "can be resumed" to "can be resummed"

---

## Round 3 · List of Changes

We thank both referees for their useful comments.

Answer to referee 1:

1- We explain in the introduction that the model we consider may be viewed as coding for the long time dynamics of the Heisenberg XX spin chain with random dephasing. We clarified the reference to [12] by indicating the appropriate section in this reference.

2- We wrote in the text that the SSEP model is recovered if one only deals with the diagonal part of the average density matrix. The relation to SSEP is pointed at the top of p.3.

3- The other comments and request (mainly typos) have been taken care of.

Answer to referee 2:

1- Although these stochastic models are clearly in the same `universality class’ (in a vague sense) as random quantum circuits, to make the connection mathematically precise is beyond the scope of this paper.

We modified the sentence in the 3rd line on p.3. to take care of this comment. See also the reply to the first referee.

2- We clarify this point and added the suggested references.

3- Thank you. Done. See also the modification in the abstract.

4- Thank you. Done.

5- One needs to be more than familiar with RMT to recognise these expectations as characteristic of RMT.

6- We clarified this point.

7- Done.

8- See answer to referee 1. The relation to SSEP is pointed at the top of p.3.

9- We think that Section 5 is interested enough by itself to be in the main text.

10- Thank you for the comment. We added a sentence to make this point. Of course, we don’t have (yet?) a proof of our conjecture.

---

## Editorial Decision

published